# Sorption of colored vs. noncolored organic matter by tidal marsh soils

**Patrick J. Neale[1], J. Patrick Megonigal[1], Maria Tzortziou[2], Elizabeth A. Canuel[3], Christina R. Pondell[3], and Hannah Morrissette[1,4]**

[1]Smithsonian Environmental Research Center, Edgewater, Maryland, USA

[2]Department of Earth & Atmospheric Science, City College of New York, New York, New York, USA

[3]`TS1`Virginia Institute of Marine Science, William & Mary, Gloucester Point, Virginia, USA

[4]`TS2`University of Maryland Center for Environmental Science, Cambridge, Maryland, USA

**Correspondence:** Patrick J. Neale (nealep@si.edu)

**Abstract.** Tidal marshes are significant sources of colored (or chromophoric) dissolved organic carbon (CDOC) to adjacent waters and, as a result, contribute substantially to their optical complexity and ultimately affect their water quality. Despite this, our mechanistic understanding of the processes that regulate the exchange and transformation of CDOC at the tidal marsh–estuarine interface remains limited. We hypothesized that tidal marsh soils regulate this exchange and transformation subject to soil mineralogy and salinity environment. To test this hypothesis, we generated initial mass sorption isotherms of CDOC and noncolored dissolved organic carbon (NCDOC) using anaerobic batch incubations of Great Dismal Swamp DOC with four tidal wetland soils, representing a range of organic carbon content ($1.77 \pm 0.12\,\%$ to $36.2 \pm 2.2\,\%$) and across four salinity treatments (0, 10, 20, and 35). CDOC sorption followed Langmuir isotherms that were similar in shape to those of total DOC, but with greater maximum sorption capacity and lower binding affinity. Like isotherms of total DOC, CDOC maximum sorption capacity increased and binding affinity decreased with greater salinity. Initial natively adsorbed colored organic carbon was low and increased with soil organic content. In contrast, NCDOC desorbed under all conditions with desorption increasing linearly with initial CDOC concentration. This suggests that for our test solutions CDOC displaced NCDOC on tidal marsh soils. Parallel factor analysis of 3-D excitation emission matrices and specific ultraviolet absorbance measurements suggested that CDOC sorption was driven primarily by the exchange of highly aromatic humic-like CDOC. Taken together, these results suggest that tidal marsh soils regulate

export and composition of CDOC depending on the complex interplay between soil mineralogy, water salinity, and CDOC vs. NCDOC composition.

## 1 Introduction

Tidal marshes are important sources to adjacent waters of colored or chromophoric dissolved organic matter (CDOM; see Appendix A Table A1 for a list of all abbreviations), the fraction of dissolved organic matter (DOM) that absorbs electromagnetic radiation across the ultraviolet (UV) and visible light spectrum (Tzortziou et al., 2008; Knobloch et al., 2022). Studies have found that CDOM composition in flooding and ebbing tidal waters differs, with more strongly colored and aromatic CDOM of higher average molecular weight and polydispersity [breadth of the molecular weight distribution]`CE1` being exported from tidal marshes into estuarine waters during ebbing tide relative to that imported into marshes during flooding tide (Tzortziou et al., 2008, 2011). This suggests that the exchange and transformation of CDOM at the tidal marsh–estuarine interface contributes substantially to the optical complexity of both estuarine and coastal waters. This complexity, in turn, can impact downstream productivity and water quality. For example, polyphenolic humic substances, an important constituent of CDOM, serve as a source of energy and nutrients to aquatic organisms (Maie et al., 2006; Coble, 2007), and their absorption of solar radiation both influences photosynthesis in aquatic envi-

ronments (Vahatalo et al., 2005) and protects aquatic organisms from harmful UV radiation (Hargreaves, 2003). Thus, in the interest of better informing tidal wetlands management, policymaking, and climate change mitigation, it is important to improve our understanding of how tidal marshes regulate chromophoric solute exchange at this vital interface.

There exists a rich body of literature focused on the sources of autochthonous wetland DOM and CDOM that includes the leaching of secondary metabolites from fresh and senescent plant material (e.g., Pinsonneault et al., 2016; Shelton et al., 2021), microbial and root exudates (Wu et al., 2012; Wei et al., 2023), and interstitial waters (e.g., Bertilsson and Jones, 2003). However, sorption processes regulating solute exchange between soils and the surrounding waters in tidal wetlands have received little consideration despite evidence of their importance in other ecological settings such as streams, marine and freshwater sediments, terrestrial soils, and non-tidal wetlands (e.g., Qualls, 2000; McKnight et al., 2002; Grybos et al., 2009; Kothawala et al., 2012). Studies have reported that soil non-crystalline iron content ([NC-Fe]) and aluminum content ([NC-Al]) play a significant role in the sorption of dissolved organic carbon (DOC), the carbon backbone of DOM, with highly aromatic moieties, carboxylic acid groups, and amino acid residues being preferentially sorbed to the soil mineral matrix (McKnight et al., 2002; Lalonde et al., 2012; Riedel et al., 2013). Concomitantly, other moieties may be preferentially desorbed, with the net exchange impacting the optical properties of the solution (Kothawala et al., 2012). Studies of solute exchange, particularly the exchange of colored solutes, are nearly absent in tidal marsh soils (Pinsonneault et al., 2021). Considering that colored and noncolored DOC have distinct properties that impact their biogeochemical cycling and influence on receiving aquatic ecosystems, the absence of studies that partition total DOC into functional categories limits our ability to forecast climate change impacts at the terrestrial–aquatic interface (Ward et al., 2020).

The salinity of tidal marsh waters, and thus their ionic strength, depends on their proximity to the estuarine–oceanic interface and ranges from freshwater to $\sim 35$. In brackish settings salinity can vary from hourly to seasonal scales in response to tidal exchange and freshwater discharge from the watershed. All coastal systems are also susceptible to long-term changes in their salinity environment resulting from rising sea levels (Wigley, 2005), changes in precipitation (Smith et al., 2005), and anthropogenic influences such as freshwater withdrawal (Li and Pennings, 2018). As ionic strength influences the structural conformation of colored (or chromophoric) dissolved organic carbon (CDOC) and can block or displace ions from exchange sites on the soil surface (Seitzinger et al., 1991; Stumm and Morgan, 2012), such changes are likely to impact the sorption of organic carbon in marsh soils and thus the quality of carbon exported to estuarine waters.

Pinsonneault et al. (2021) previously demonstrated that sorption of total dissolved organic carbon at the tidal marsh–estuarine interface is primarily regulated by soil mineralogy and salinity environment. Here, we analyze sorption processes in more detail by considering separately the exchange of CDOC and noncolored dissolved organic carbon (NC-DOC) with tidal marsh soils. As in our previous work, we examine these exchanges across tidal marshes that vary in salinity regime (freshwater to saline) and soil organic carbon content (mineral-dominated soils to highly organic soils). Overall, the objectives of this study were to (1) quantify the exchange of CDOC and NCDOC between the dissolved phase and tidal marsh soils, (2) identify how these exchanges are affected by salinity and soil characteristics, and (3) quantify the effect of these exchanges on the optical characteristics and inferred molecular composition of CDOM exported from tidal marshes to coastal waters.

## 2 Methods

### 2.1 Soils

Between 5 March and 15 March 2018, we collected 40 cm deep soil cores from four tidal marshes in the Chesapeake Bay area of the mid-Atlantic United States that ranged in salinity from freshwater to $\sim 32$ and in soil total organic carbon (TOC) content from mineral-dominated ($\sim 2\%$ TOC) to highly organic (36% TOC). These four wetlands were the Smithsonian Environmental Research Center's Global Change Research Wetland (GCReW) (located in Kirkpatrick Marsh, 36°53′ N, 76°33′ W), the Jug Bay Wetlands Sanctuary (38°46′ N, 76°42′ W), Taskinas Marsh located in the York River State Park (37°25′ N, 76°43′ W), and Wachapreague Marsh (37°32′ N, 75°41′ W). Tidal marsh surface and soil core characteristics were reported by Pinsonneault et al. (2021) and for reference are summarized in Appendix B Table B1. Descriptions of all sampling and analytical methods are provided by Pinsonneault et al. (2021). Additional site and sediment descriptions are provided by Pondell and Canuel (2022).

### 2.2 Dissolved organic carbon standards

To provide a highly concentrated standard of natural CDOM that could be diluted to a range of concentrations for incubations with tidal marsh soils, we collected highly colored surface waters from the Great Dismal Swamp National Wildlife Refuge (GDS), a non-tidal forested wetland located on the coastal plain of southeastern Virginia and northeastern North Carolina, USA (36°37′ N, 76°28′ W). We sampled a non-tidal system to have a consistent source of strongly absorbing, terrestrial DOM that could be used for a series of experiments. In April 2018, we collected approximately 115 L of surface water in acid-washed carboys from the Jericho Ditch (36°41′45.03″ N, 76°30′28.16″ W). This sample was concen-

trated by reverse osmosis (RO) using a Growonix GX600 RO system fitted with both pleated and spun sediment filters before being filtered through a 20 µm pore size Whatman Poly-cap 75 HD disposable filter capsule followed by a 0.2 µm pore size Whatman Polycap 36 TC polyethersulfone membrane capsule. The resulting filtered DOC concentrate was then treated with 1 mM sodium azide (NaN$_3$) and a microbial inhibitor, and DOC concentration ([DOC]) was measured with a Shimadzu TOC-L using high-temperature combustion (reproducibility 1.5 % according to manufacturer). This concentrate yielded a [DOC] of 217 mg-DOC L$^{-1}$, a specific conductivity of 180 µS cm$^{-1}$ corresponding to a salinity of 0.08, and a pH of 4.40. The treated concentrate was divided into four sub-stocks that we amended with Instant Ocean aquarium salt (a synthetic sea salt) to produce four salinity treatments: 0 (no instant ocean added), 10, 20, and 35 on the practical salinity scale (no units). The typical ionic composition of Instant Ocean is reported by Christy and Dickman (2002).

## 2.3 Batch isotherm incubations

Seven isotherm standards, ranging in [DOC] from 0 to $\sim 200$ mg L$^{-1}$, were prepared by diluting the DOC treatment stocks with a $\sim 0$ mg-DOC L$^{-1}$ solution (hereafter referred to as the dilutant) of similar pH, salinity, and specific conductivity prepared using ultrapure water, Instant Ocean aquarium salt, and 1 mM NaN$_3$. Isotherm standards were freshly prepared for each batch incubation.

Based on methods described by Kothawala et al. (2008) we carried out batch incubations under a 95 % nitrogen (N$_2$) and 5 % hydrogen atmosphere in a Coy Laboratory Products anaerobic chamber using aqueous samples and solutions degassed with N$_2$. Soil samples were incubated in triplicate at room temperature ($\sim 22$°C) using a 1 : 30 ratio of freeze-dried soil mass (g) to isotherm standard volume (mL) for 24 h in the dark on a horizontal shaker (70 RPM). We then centrifuged the incubated soil slurries at 4700 g for 1 h and measured the salinity and pH of the supernatant using a WTW Multi 340i probe and a Thermo Orion 3 Star pH meter, respectively. Finally, the supernatant was syringe-filtered using disposable 0.45 µm Millex filter cartridges for DOC and spectral analyses. We confirmed the lack of DOC precipitation in the absence of soil by conducting test incubations without soil at 12.5 and 285 mg-DOC L$^{-1}$ across the four salinity treatments (data not shown).

## 2.4 Colored and noncolored dissolved organic carbon

Due to the highly colored nature of the Great Dismal Swamp DOC, we diluted subsamples of pre-incubation standards by a factor of 10 using dilutant from the same salinity treatment while we diluted the filtered post-incubation supernatant by the same factor using ultrapure water + NaCl (Sigma-Aldrich, 99.5 % purity) to match sample salinity.

All solutions were degassed with N$_2$, then dilutions were performed, and cuvettes loaded and sealed in an anaerobic chamber. We performed absorbance scans at 2 nm intervals (270–750 nm) for all replicates using a Thermo Scientific Evolution 220 UV–Vis spectrophotometer. Specific ultraviolet absorbance at 280 nm (SUVA$_{280}$), an indicator of DOC aromaticity, was calculated from these data by dividing decadic sample absorbance at 280 nm by the DOC concentration ([DOC]) (Hansen et al., 2016 TS3). We then generated three-dimensional excitation–emission matrices (EEMs) using a Horiba Jobin Yvon FluoroMax-3 spectrofluorometer (for sample replicate A TS4 only) at 5 nm intervals (250–600 nm) for excitation and 2 nm intervals (250–600 nm) for emission. Fluorescence spectra were corrected for inner-filter effect and Raman scattering using the drEEM toolbox version 0.2.0 (Murphy et al., 2013 TS5) in MATLAB (v. 2017b). Parallel factor analysis (PARAFAC) was used to deconstruct the fluorescence signal into underlying fluorescence components, or fluorophores, that relate to differences in DOC composition (Murphy et al., 2010 TS6; Lapierre and del Giorgio, 2014).

Although all incubations started with 100 % of the DOC as GDS organic carbon (apart from trace additions from the dilutant), adsorption–desorption interactions with the soil changed the composition of organic carbon in the dissolved phase, thus changing the spectral characteristics of absorbance and fluorescence. To better understand how adsorption–desorption reactions change DOC composition, we developed a partitioning approach that considers the total DOC pool composed of two components: colored and noncolored. The colored component (CDOC) is defined as the fraction of DOC that varies linearly with absorbance. Operationally, this was estimated for each incubation using a linear regression of DOC as a function of absorbance at 355 nm (MATLAB fitlm), following the methods of Clark et al. (2019). For the purposes of this analysis, any DOC that absorbed UV only at wavelengths < 355 nm was considered noncolored. The data analysis conducted to convert $a355$ to CDOC (mg L$^{-1}$) is presented in the results section. Once CDOC was determined, the noncolored fraction (NCDOC) was estimated by difference with total DOC, which is here designated as "TDOC", following Eq. (1):

$$[NCDOC] = [TDOC] - [CDOC]. \tag{1}$$

## 2.5 CDOC sorption

To quantify CDOC exchange with tidal marsh soils, we used a similar approach as described previously for TDOC sorption parameters (Pinsonneault et al., 2021). CDOC sorption was estimated by least-squares non-linear fitting using the traditional Langmuir isotherm, except in this case the net sorption of CDOC was considered dependent on the *initial* instead of the final concentration of the absorbent:

$$\Delta[COC]_{i-f} = \frac{(Q_{Cmax} \cdot K_C \cdot [CDOC]_i)}{(1 + (K_C \text{ TS7} \cdot [CDOC]_i))} - C_{C0}, \tag{2}$$

where $\Delta[COC]_{i-f}$ is the quantity of colored organic carbon (COC) adsorbed in mg-C g$^{-1}$, $[CDOC]_i$ is the initial CDOC concentration in mg-DOC L$^{-1}$, $K_C$ is the CDOC binding affinity in L mg$^{-1}$, $Q_{Cmax}$ is the soil maximum adsorption capacity for COC in mg-C g$^{-1}$, and $C_{C0}$ is the initial natively sorbed COC in mg-C g$^{-1}$ (cf. Kothawala et al., 2008). The COC adsorbed is based on the difference between $[CDOC]_i$ and $[CDOC]_f$, the CDOC in solution at the end of the incubation. $K_C$, $Q_{Cmax}$, and $C_{C0}$ for each incubation were estimated by fitting Eq. (2) to the results of each experiment using nonlinear regression (MATLAB fitnlm).

The null point (NP) is defined as the concentration at which there is no net removal (adsorption) or release (desorption) of CDOC from solution, the biogeochemical significance of which is a sorptive equilibrium between the soil mineral surfaces and the aqueous phase. It is derived by setting Eq. (2) to zero and solving for NP concentration of $[CDOC]_i$:

$$NP = \frac{C_{C0}}{(Q_{Cmax} - C_{C0}) \cdot K_C}. \tag{3}$$

We chose to model CDOC adsorption in this analysis using the initial mass (IM) Langmuir isotherm since we use the same dependent variable to model the dynamics of NCDOC. For reasons to be shown in results, NCDOC dynamics can be accounted for using a linear, desorption-only, IM isotherm:

$$\Delta[NCOC]_{i-f} = m_N \cdot [CDOC]_i - C_{N0}, \tag{4}$$

where $\Delta[NCOC]_{i-f}$ is the difference between initial and final adsorbed noncolored organic carbon mg-C g$^{-1}$, $m_N$ (L g$^{-1}$) is a desorption coefficient, and $C_{N0}$ is the native adsorbed NCOC. Implicit in this equation is the assumption that soil sorption of COC displaces NCOC, releasing it into solution. The slope of the NCDOC isotherm can be compared to the initial slope of the CDOC isotherm ($m_C = K_C \cdot Q_{Cmax}$, L g$^{-1}$). The ratio $m_N/m_C$ is termed the displacement coefficient, the proportion of CDOC adsorption associated with NCOC desorption.

IM isotherms are advantageous for use in biogeochemical models since the result is a function of the initial, not final condition (Kothawala et al., 2008). While equilibrium Langmuir theory leads to the traditional rather than IM isotherm, both formulations are equally effective at explaining soil DOC adsorption–desorption (Kothawala et al., 2008). The parameters for the fits of the IM Langmuir isotherms to the TDOC data previously presented by Pinsonneault et al. (2021) are provided in Appendix B Table B2.

## 3 Results

### 3.1 Colored and noncolored dissolved organic carbon content

Linear regressions of TDOC on absorbance at 355 nm were performed for each set of initial standards and final filtrates.

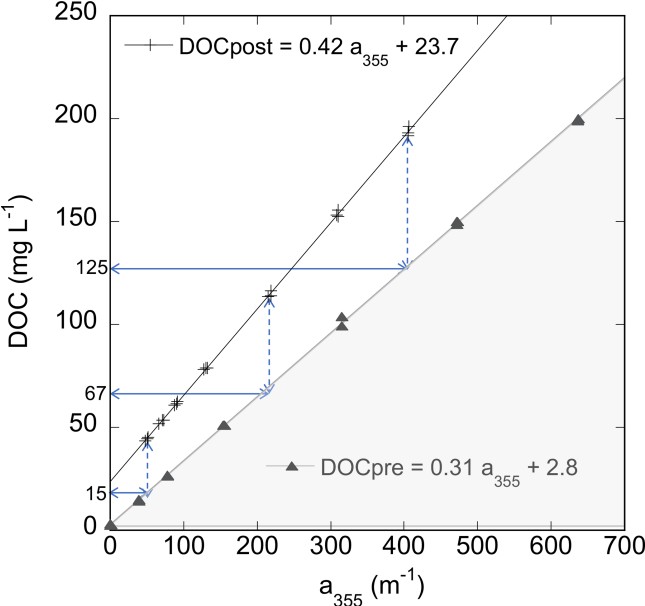

**Figure 1.** Scatterplot of DOC content vs. absorption coefficient at 355 nm in pre-incubation (triangles) and post-incubation (+ symbols) treatments in triplicate from the Kirkpatrick soil, $S = 10$, isotherm experiment with fitted linear regressions. The dashed arrows indicate the difference in DOC between the regression lines for selected post-incubation absorbances; the solid arrows identify the DOC of the pre-incubation line at these absorbances.

Figure 1 shows a typical set of regression results for Kirkpatrick Marsh, treatment with salinity ($S$) of 10, and all results are provided in a supplemental file. Note the offset between the post-incubation and the pre-incubation regression lines. For any given absorbance there was more DOC in solution after incubation than before due to an increase in NCDOC. The CDOC content corresponding to a given absorbance is defined by the pre-incubation regression line (shaded triangles, in Fig. 1). To illustrate the process, dashed arrows in Fig. 1 project the absorbance in each post-incubation treatment down to the pre-incubation fitted regression line. This point (minus the intercept) is the inferred CDOC content as indicated by the solid arrows that project to the $y$ axis (DOC). As examples, the three post-incubation solutions with TDOC = 44, 115, and 194 mg L$^{-1}$ project to CDOC contents of 15, 67, and 125 mg L$^{-1}$, respectively. The length of the dashed arrow represents the remainder between CDOC and TDOC in the post-incubation solution that is the noncolored DOC (NCDOC, dashed arrows) desorbed during the incubation. In this example, NCDOC rises from $\sim 29$ to 68 mg L$^{-1}$ in the post-incubation solutions with the lowest and highest concentrations of TDOC (bottom and top dashed arrows, respectively).

The shift from pre-incubation to post-incubation regression lines shown in Fig. 1 happened in a similar way in every experiment. Like Fig. 1, in each experiment both the

pre- and post-incubation solutions showed tight linear regressions with positive intercepts ($r^2 > 0.996$, $n = 21$, Table 1), with higher slopes and intercepts for the post-incubation solutions. There were small intercepts ($\sim 2\,\mathrm{mg\,L^{-1}}$) for the initial standards stemming from a trace component of TDOC in the dilutant. Because the dilutant had no absorbance at 355 nm, this was considered NCDOC. However, the dilutant did absorb at wavelengths $< 300\,\mathrm{nm}$ (Appendix B Fig. B1). This absorbance biases the spectral slope estimates of CDOC for treatments that have been diluted. To minimize such bias, spectral slope estimates (Sect. 3.3) are based on the average of the three highest standards which had the least dilution ($< 12\,\%$ dilution).

Incubation of the standards with tidal marsh soils dramatically increased the contribution of NCDOC to the solution phase. This is illustrated in Fig. 2 for the example dataset presented in Fig. 1. The pre-incubation standards used the highly colored GDS solution to which a background level of NCDOC was introduced during dilution. This is the DOC in the "0" standard ($3\,\mathrm{mg\,L^{-1}}$ in Fig. 2). For the other standards, CDOC made up more than 80 % of the initial DOC. Except for the "0" standard, the proportion of CDOC was much lower and NCDOC higher after incubation. NCDOC was desorbed in every case and made up more than 50 % of the DOC in the four lowest standards and 35 %–40 % at the higher standards. Conversely, CDOC decreased to between 34 %–65 % (from the lowest to highest standard) of the post-incubation DOC. In this example, there was only net desorption of CDOC in the lowest three standards, transitioning to net adsorption for the four highest standards (compare DOC values in colored bars, Fig. 2). This contrasts with the net desorption of total DOC for all except the highest standard (compare DOC values on $x$ axes). While the transition point from desorption to adsorption varied among the experiments, this general pattern of always net *desorption* for NCDOC and mostly net *adsorption* for CDOC was observed for all experimental soils and treatments.

## 3.2 CDOC sorption isotherms

The sorption isotherm plots of CDOC for all four marsh soils are presented in Fig. 3 (filled symbols) and the sorption characteristics estimated by fitting the IM Langmuir isotherm (Eq. 2) to the data shown in Table 2. The Langmuir equation provided a good fit to the isotherm data for all experiments, except that reliable estimates of soil maximum adsorption capacity and binding affinity could not be obtained for Kirkpatrick and Taskinas marshes for $S = 35$ (Table 2). These isotherms were close to linear over the measured range, as was the case for the TDOC isotherms (Pinsonneault et al., 2021, Appendix B). The minimum $r^2$ of the nonlinear regressions for all soils across the four salinity treatments was 0.893, and the mean $r^2$ was $0.975 \pm 0.028$ ($n = 21$). The isotherms of CDOC are similar in shape to the TDOC isotherms, both the traditional Langmuir isotherms

previously presented by Pinsonneault et al. (2021) and the IM Langmuir isotherms refitted to their data (Appendix B Fig. B2). As for these TDOC isotherms (see Appendix Table B2), the maximum soil sorption capacity for CDOC generally increased with salinity, while the CDOC binding affinity tended to decrease with salinity (Table 2).

## 3.3 NCDOC desorption isotherms

The isotherms for NCDOC (Fig. 3, open symbols) contrasted sharply with those of CDOC. The difference between initial and final adsorbed noncolored organic carbon ($\Delta[\mathrm{NCOC}]_{i-f}$, $\mathrm{mg\,g^{-1}}$) was always negative (net desorption), and the desorption magnitude was linearly related to initial CDOC concentration by the desorption coefficient, $m_N$ (Eq. 4, fitted parameters in Table 3). The displacement coefficient, the ratio between $m_N$ and the initial slope of the CDOC isotherm ($m_C$), varied from 0.12 to 0.44 (Table 3). The displacement coefficient shows that between 12 %–44 % of the adsorption of CDOC is associated with the desorption of NCOC.

## 3.4 Relationship of isotherm parameters to soil characteristics

Although the parameter trends of CDOC and TDOC isotherms were similar, the magnitudes of the parameters were quite different. These differences are illustrated in an example set of fits for the Kirkpatrick, $S = 10$ experiment (data in Fig. 1) for both TDOC and CDOC IM isotherms (Fig. 4). The $y$ intercept represents $C_0$, the native adsorbed organic carbon. It is clear that the absolute value of the intercept was much smaller in magnitude for the CDOC (solid circles) vs. the TDOC ("x" symbols) isotherms, e.g., 0.5 vs. $1.5\,\mathrm{mg\,g^{-1}}$ in Fig. 4. The $C_0$ for CDOC (i.e., $C_{C0}$) was lower because the $C_0$ for TDOC isotherms had a large component of native-adsorbed *noncolored* organic carbon. The differences between TDOC and CDOC $C_0$s are summarized in Table 4. Native adsorbed COC as a proportion of total organic carbon (TOC) varied among treatments but averaged around 39 % for the more organic-rich Taskinas and Kirkpatrick soils and only 6 % for the more mineral Jug Bay and Wachapreague soils. The $C_{C0}$ decreased with salinity for Taskinas and Kirkpatrick soils and in general increased as a function of soil % organic matter with Kirkpatrick soils having the highest $C_{C0}$ (Fig. 5a). The noncolored $C_{N0}$ was estimated from the NCDOC isotherm (Table 3) or, equivalently, the difference between colored and total native adsorbed carbon (Table 4). The two estimates were statistically identical (compare Tables 3 and 4). In contrast with $C_{C0}$, $C_{N0}$ was mainly a function of soil-specific surface area (Fig. 5b) as reported in Appendix B Table B1. Desorption of NCOC increased (i.e., $m_n$ more negative) with salinity (Fig. 5c), with desorption coefficients similar for brackish and fresh tidal marsh soils. Both $C_{N0}$ and $m_n$ were consis-

**Table 1.** Results of linear regression fits for DOC (mg L$^{-1}$) vs. absorbance at 355 nm (m$^{-1}$) for both pre-incubation standards (left side) and post-incubation filtrates (right side). Estimates are stated $\pm$ SE. A separate regression ($n = 21$) was estimated for each soil and salinity treatment; plots of the data and estimated regression lines are provided in the supplemental file. CE2

| Salinity | Kirkpatrick | Taskinas | Jug Bay | Wachapreague | Kirkpatrick | Taskinas | Jug Bay | Wachapreague |
|---|---|---|---|---|---|---|---|---|
| Pre-incubation standards | | | | | Post-incubation filtrates | | | |
| DOC intercept (mg L$^{-1}$) | | | | | DOC intercept (mg L$^{-1}$) | | | |
| 0 | $1.71 \pm 0.37$ | $1.02 \pm 0.34$ | $1.98 \pm 0.68$ | $1.56 \pm 0.51$ | $22.75 \pm 1.06$ | $23.87 \pm 0.88$ | $29.88 \pm 0.90$ | $7.93 \pm 0.44$ |
| 10 | $2.82 \pm 0.42$ | $2.42 \pm 0.28$ | $0.40 \pm 0.59$ | $1.82 \pm 0.48$ | $23.67 \pm 0.37$ | $34.68 \pm 0.78$ | $35.17 \pm 1.24$ | $11.72 \pm 0.84$ |
| 20 | $1.21 \pm 0.23$ | $0.11 \pm 0.37$ | $2.07 \pm 0.30$ | $1.31 \pm 0.34$ | $25.99 \pm 0.96$ | $31.12 \pm 1.09$ | $32.51 \pm 1.63$ | $9.28 \pm 0.67$ |
| 35 | $1.93 \pm 0.52$ | $1.07 \pm 0.56$ | $2.26 \pm 0.70$ | $0.74 \pm 0.47$ | $25.73 \pm 0.74$ | $32.79 \pm 0.81$ | $33.27 \pm 0.97$ | $7.95 \pm 0.43$ |
| DOC slope (g m$^2$) | | | | | DOC slope (g m$^2$) | | | |
| 0 | $0.322 \pm 0.001$ | $0.313 \pm 0.001$ | $0.304 \pm 0.002$ | $0.309 \pm 0.002$ | $0.383 \pm 0.003$ | $0.340 \pm 0.003$ | $0.344 \pm 0.004$ | $0.321 \pm 0.001$ |
| 10 | $0.311 \pm 0.001$ | $0.310 \pm 0.001$ | $0.344 \pm 0.002$ | $0.315 \pm 0.001$ | $0.419 \pm 0.002$ | $0.411 \pm 0.004$ | $0.641 \pm 0.016$ | $0.327 \pm 0.003$ |
| 20 | $0.303 \pm 0.001$ | $0.313 \pm 0.001$ | $0.310 \pm 0.001$ | $0.307 \pm 0.001$ | $0.435 \pm 0.005$ | $0.443 \pm 0.006$ | $0.636 \pm 0.023$ | $0.346 \pm 0.003$ |
| 35 | $0.312 \pm 0.002$ | $0.292 \pm 0.002$ | $0.291 \pm 0.002$ | $0.289 \pm 0.001$ | $0.480 \pm 0.004$ | $0.440 \pm 0.005$ | $0.624 \pm 0.015$ | $0.353 \pm 0.002$ |
| $r^2$ | | | | | $r^2$ | | | |
| 0 | 1.000 | 1.000 | 0.999 | 1.000 | 0.998 | 0.999 | 0.997 | 1.000 |
| 10 | 1.000 | 1.000 | 0.999 | 1.000 | 1.000 | 0.998 | 0.988 | 0.998 |
| 20 | 1.000 | 1.000 | 1.000 | 1.000 | 0.997 | 0.996 | 0.976 | 0.999 |
| 35 | 1.000 | 0.999 | 0.999 | 1.000 | 0.998 | 0.998 | 0.989 | 1.000 |

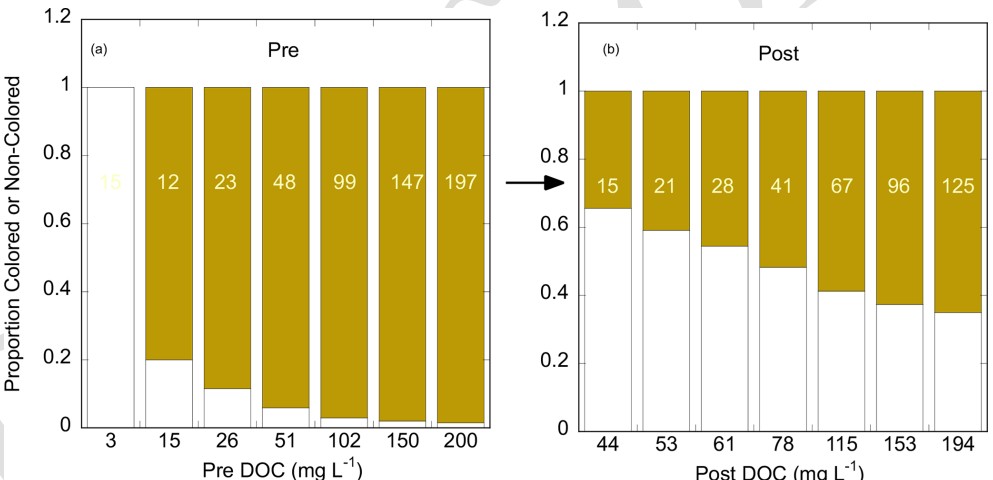

**Figure 2.** Proportions of noncolored (white bars) and colored (brown bars) DOC in **(a)** the pre-incubation ("Pre") and **(b)** post-incubation ("Post") solution phases of an experimental incubation of Kirkpatrick tidal marsh soil, $S = 10$ (cf. Fig. 1). $x$ axes are annotated with the total DOC for each of the seven standards; annotations in the colored bars are the inferred concentrations of CDOC (mg L$^{-1}$) in the pre-incubation **(a)** and post-incubation **(b)** solutions.

tently smaller (i.e., for $m_n$, less negative) in incubations of the sandy Wachapreague soils.

The maximum sorption capacity of CDOC ($Q_{Cmax}$, mg g$^{-1}$) was closely related to that of the TDOC IM isotherm but was about 4 times higher (Fig. 6a). The $Q_{Tmax}$ for TDOC was lower because the adsorption of CDOC is partially compensated by desorption of NCOC. The CDOC isotherms reveal that tidal marsh soils have a large capacity to adsorb COC, especially at high salinity. The larger $Q_{Cmax}$ is coupled to a smaller binding affinity, ($K_C$, L mg$^{-1}$),

which determines how quickly adsorption approaches saturation as CDOC$_i$ increases. As can be seen in Fig. 3, the approach to saturation is distinctly slower for the CDOC vs. TDOC isotherm. Overall, $K_C$ is about half of $K_T$ for TDOC (Fig. 6b). Indeed (as already mentioned), over the range of experimental treatments the CDOC IM isotherm is close to linear, though curvature is more apparent in some treatments (e.g., Jug Bay, 0 in Fig. 2).

The much smaller $C_{C0}$ for CDOC also translates to a much lower threshold concentration of CDOC$_i$ (compared

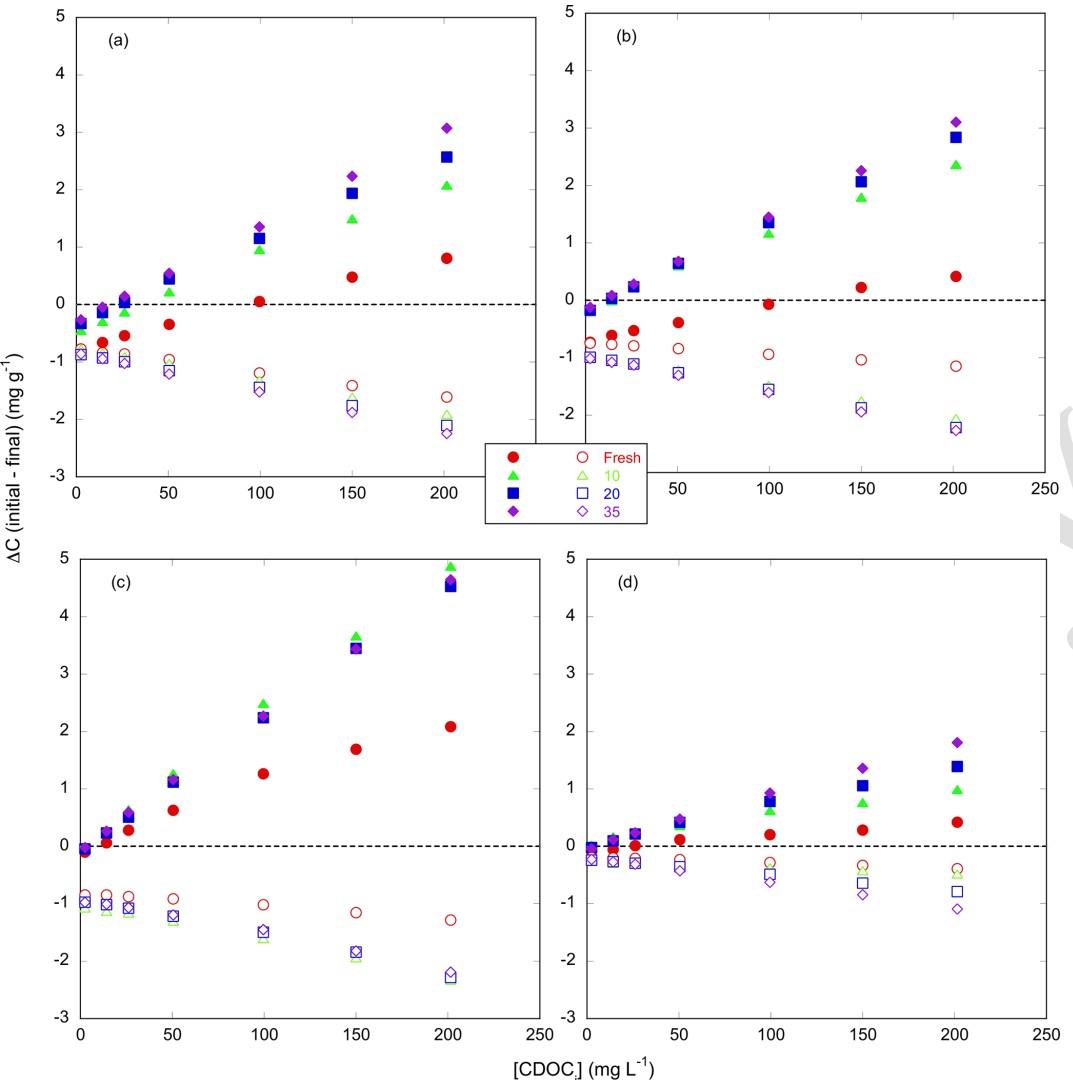

**Figure 3.** Isotherms of adsorption–desorption of CDOC (filled shapes) and NCDOC (open shapes) for tidal marsh soils. Net adsorption of organic carbon based on the difference between initial and final concentrations of soil organic carbon ($\Delta C$, mg C g-soil$^{-1}$) as a function of initial concentration of CDOC ([CDOC]$_i$, mg L$^{-1}$). Dashed line indicates no net change, and negative values indicate net desorption. Error bars show standard deviation of triplicate incubations; in most cases error bars are smaller than the symbol. Incubations used soils from **(a)** Kirkpatrick, **(b)** Taskinas, **(c)** Jug Bay, and **(d)** Wachapreague marshes. Red circles, green triangles, blue squares, and purple diamonds denote salinities of 0, 10, 20, and 35, respectively.

to TDOC$_i$) above which there is net adsorption (null point, NP, Eq. 3). The estimated *NP*s (Table 2) varied in the range of 0–32 mg L$^{-1}$ CDOC, except for values of $\sim 100$ mg L$^{-1}$ for the Kirkpatrick and Taskinas fresh treatments. For Wachapreague and Jug Bay, the NP was $< 10$ mg L$^{-1}$ except for the Wachapreague fresh treatment for which the NP was 23.5 mg L$^{-1}$. These are lower by almost an order of magnitude than the null points estimated for TDOC in these soils (Pinsonneault et al., 2021, and Appendix B Table B2).

## 3.5 DOC composition

The primary change in DOC composition during incubation was a decrease in specific absorbance, which resulted from the addition of NCDOC during the incubation. Because specific absorbance at 355 nm is the inverse of the slope of the DOC vs. absorbance plots (Fig. 1, Table 1), the increase in slope (g m$^{-2}$) across all the incubations translates to a decrease in specific absorbance ($a \cdot$ TS11 (355), m$^2$ g$^{-1}$). Similarly, the specific absorbance at 280 nm (SUVA280) and all other wavelengths decreased in all treatments (data not shown).

**Table 2.** Parameter estimates $\pm$ SE of sorption characteristics of CDOC derived from the initial mass Langmuir isotherm by salinity treatment. $Q_{\mathrm{Cmax}}$, $C_{\mathrm{C0}}$, and $K_{\mathrm{C}}$ refer to the soil maximum sorption capacity, native adsorbed colored soil carbon, and the CDOC binding affinity, respectively. Null point (NP) was estimated using fitted isotherm parameters and Eq. (3). $Q_{\mathrm{Cmax}}$ and $K_{\mathrm{C}}$ for Kirkpatrick and Taskinas for $S = 35$ incubations were estimated by extrapolation of the salinity trend, and no SE is reported. NP was not estimated using extrapolated values. * The fitted estimate of $C_{\mathrm{C0}}$ for this treatment was negative, considered to be 0 for purposes of the NP estimate.

| Variable/salinity | Kirkpatrick | Taskinas | Jug Bay | Wachapreague |
|---|---|---|---|---|
| $Q_{\mathrm{Cmax}}$ (m g$^{-1}$) | | | | |
| 0 | $9.2 \pm 4.3$ | $4.0 \pm 0.9$ | $5.4 \pm 0.5$ | $1.1 \pm 0.3$ |
| 10 | $16.8 \pm 2.9$ | $12.6 \pm 2.0$ | $47.9 \pm 5.9$ | $2.4 \pm 0.4$ |
| 20 | $36.0 \pm 12.3$ | $40.8 \pm 17.1$ | $96.4 \pm 27.1$ | $5.4 \pm 0.9$ |
| 35 | $(54.2)$ | $(65.2)$ | $78.9 \pm 21.3$ | $14.1 \pm 4.6$ |
| $C_{\mathrm{C0}}$ (mg g$^{-1}$) | | | | |
| 0 | $0.80 \pm 0.03$ | $0.73 \pm 0.02$ | $0.15 \pm 0.03$ | $0.09 \pm 0.02$ |
| 10 | $0.47 \pm 0.01$ | $0.17 \pm 0.02$ | $0.04 \pm 0.01$ | $-0.04 \pm 0.02$ |
| 20 | $0.34 \pm 0.01$ | $0.17 \pm 0.02$ | $0.06 \pm 0.01$ | $0.02 \pm 0.01$ |
| 35 | $0.28 \pm 0.01$ | $0.13 \pm 0.01$ | $0.04 \pm 0.01$ | $0.03 \pm 0.02$ |
| $K_{\mathrm{C}}$ (L mg$^{-1}$) $\times 10^{-3}$ | | | | |
| 0 | $1.02 \pm 0.58$ | $2.05 \pm 0.68$ | $3.64 \pm 0.56$ | $3.71 \pm 1.55$ |
| 10 | $0.91 \pm 0.19$ | $1.26 \pm 0.24$ | $0.52 \pm 0.07$ | $3.22 \pm 0.86$ |
| 20 | $0.45 \pm 0.17$ | $0.39 \pm 0.18$ | $0.25 \pm 0.07$ | $1.73 \pm 0.40$ |
| 35 | $(0.12)$ | $(0.13)$ | $0.32 \pm 0.09$ | $0.76 \pm 0.29$ |
| NP (mg L$^{-1}$) | | | | |
| 0 | 92.9 | 109.7 | 8.1 | 23.5 |
| 10 | 31.5 | 10.9 | 1.7 | 0.0* |
| 20 | 20.7 | 10.8 | 2.6 | 2.0 |
| 35 | n/a TS8 | n/a | 1.6 | 3.0 |
| $r^2$ | | | | |
| 0 | 0.918 | 0.958 | 0.988 | 0.893 |
| 10 | 0.988 | 0.982 | 0.995 | 0.975 |
| 20 | 0.990 | 0.985 | 0.996 | 0.980 |
| 35 | 0.994 | 0.981 | 0.992 | 0.983 |

Given that spectral characteristics of CDOM absorbance and fluorescence are used to infer sources, sinks, reactivity, and other biogeochemical processes (Chin et al., 2002; Helms et al., 2008; Tzortziou et al., 2008), we examined how spectral properties of CDOC were affected by sorption processes. In addition to the "dilution effect" of desorbed NC-DOC on DOC quality, there were small shifts in the spectral properties of CDOM absorbance during the incubations. The GDS-derived standards had an average ($\pm$ SD) spectral slope of $0.014 \pm 0.0004$ nm$^{-1}$ and slope ratio of $0.789 \pm 0.012$ (Table 5). The slope ratio ($S_{\mathrm{R}}$) falls in the range of that previously reported for GDS CDOM ($S_{\mathrm{R}} = 0.69$–$0.84$; Helms et al., 2008). The spectral slope of CDOM after incubation was unchanged for Kirkpatrick but increased slightly for the other marsh soils to an average of $0.015 \pm 0.001$ nm$^{-1}$. Interestingly, the average increase was slightly greater for $S = 20$ (0.017) and less for $S = 35$ (0.014). Slope ratio de-

creased on average to $0.75 \pm 0.030$. Despite this general decrease in slope ratio, trends differed among the soils in the 275 to 295 nm slope ($S_{275--295}$), which is inversely related to the lignin-derived content of CDOM (Fichot and Benner, 2012 TS12). For the organic-rich soils of Kirkpatrick marsh, incubation resulted in a slight decrease in $S_{275--295}$, averaging $0.0127$ nm$^{-1}$ for the standard and $0.0125$ nm$^{-1}$ for post-incubation CDOM. However, for the other soils the magnitude of $S_{275--295}$ increased to $0.0143 \pm 0.001$ nm$^{-1}$, indicating a decrease in lignin-derived CDOM. Within each of the standards and the post-incubation data for each soil, the lowest slope and $S_{275--295}$ occurred at the highest salinity (Table 5).

Given these shifts in CDOM composition during incubation, we used parallel factor analysis of excitation–emission matrices (PARAFAC) to obtain more information on the molecular composition. PARAFAC validated six fluores-

**Table 3.** Parameter estimates $\pm$ SE for desorption isotherms of NCDOC based on the linear equation $\Delta[\text{NCOC}]_{i-f} = m_N \cdot$ TS9 $\text{CDOC}_i - C_{N0}$, where $\Delta[\text{NCOC}]_{i-f}$ is difference between initial and final adsorbed noncolored organic carbon (mg g$^{-1}$), $m_N$ (L g$^{-1}$) is the desorption coefficient for $[\text{CDOC}]_i$, the initial concentration of colored dissolved organic carbon (mg L$^{-1}$), and $C_{N0}$ is the native adsorbed NCOC (mg g$^{-1}$). $r^2$ values for all fits ($n = 21$, per experiment) were $> 0.99$. Displacement coefficient is the proportion of CDOC adsorption associated with NCOC desorption.

| Variable/salinity | Kirkpatrick | Taskinas | Jug Bay | Wachapreague |
|---|---|---|---|---|
| $C_{N0}$ (mg g$^{-1}$) | | | | |
| 0 | $0.77 \pm 0.01$ | $0.74 \pm 0.00$ | $0.82 \pm 0.01$ | $0.19 \pm 0.00$ |
| 10 | $0.77 \pm 0.01$ | $0.98 \pm 0.01$ | $1.03 \pm 0.01$ | $0.27 \pm 0.01$ |
| 20 | $0.85 \pm 0.01$ | $0.96 \pm 0.01$ | $0.92 \pm 0.02$ | $0.23 \pm 0.00$ |
| 35 | $0.86 \pm 0.01$ | $0.99 \pm 0.01$ | $0.92 \pm 0.01$ | $0.21 \pm 0.00$ |
| $m_N$ (L g$^{-1}$) $\times 10^{-3}$ | | | | |
| 0 | $-4.12 \pm 0.05$ | $-2.00 \pm 0.02$ | $-2.33 \pm 0.07$ | $-0.99 \pm 0.02$ |
| 10 | $-5.75 \pm 0.06$ | $-5.30 \pm 0.07$ | $-5.75 \pm 0.11$ | $-1.08 \pm 0.05$ |
| 20 | $-6.32 \pm 0.07$ | $-6.09 \pm 0.08$ | $-6.55 \pm 0.18$ | $-2.80 \pm 0.03$ |
| 35 | $-6.86 \pm 0.05$ | $-6.48 \pm 0.10$ | $-6.30 \pm 0.13$ | $-4.43 \pm 0.04$ |
| Displacement coefficient (rel.) | | | | |
| 0 | 0.44 | 0.24 | 0.23 | 0.12 |
| 10 | 0.38 | 0.33 | 0.14 | 0.23 |
| 20 | 0.39 | 0.38 | 0.30 | 0.27 |
| 35 | 0.41 | 0.40 | 0.41 | 0.25 |

**Table 4.** Comparison of native sorbed organic carbon estimates for total ($C_0$), colored ($C_{C0}$), and noncolored organic carbon ($C_{N0}$), based on fitted Langmuir isotherms. $C_{N0}$ was estimated by the difference $C_0 - -C_{C0}$. The $C_{N0}$ based on this difference is statistically identical to the $C_{N0}$ of the desorption isotherm (Table 3), $r = 0.99$. TS10 * The fitted estimate of $C_{C0}$ for this treatment was negative, considered to be 0 for purposes of the comparison.

| | Total-$C_0$ | Colored-$C_{C0}$ | Noncolored-$C_{N0}$ | Colored % of total |
|---|---|---|---|---|
| Kirkpatrick | | | | |
| 0 | 1.60 | 0.80 | 0.80 | 50 % |
| 10 | 1.26 | 0.47 | 0.79 | 37 % |
| 20 | 1.23 | 0.34 | 0.89 | 27 % |
| 35 | 1.15 | 0.28 | 0.87 | 25 % |
| Taskinas | | | | |
| 0 | 1.49 | 0.73 | 0.76 | 49 % |
| 10 | 1.17 | 0.17 | 1.00 | 14 % |
| 20 | 1.15 | 0.17 | 0.98 | 15 % |
| 35 | 1.13 | 0.13 | 1.00 | 12 % |
| Jug Bay | | | | |
| 0 | 1.04 | 0.16 | 0.88 | 15 % |
| 10 | 1.12 | 0.04 | 1.08 | 4 % |
| 20 | 1.10 | 0.06 | 1.04 | 6 % |
| 35 | 1.06 | 0.04 | 1.02 | 4 % |
| Wachapreague | | | | |
| 0 | 0.34 | 0.09 | 0.25 | 27 % |
| 10 | 0.22 | 0* | 0.22 | 0 % |
| 20 | 0.30 | 0.02 | 0.28 | 6 % |
| 35 | 0.26 | 0.03 | 0.23 | 12 % |

**Table 5.** Average spectral properties of CDOM for solutions added to (Pre) or resulting from incubation of tidal marsh soils. Tabulated are the full spectral slope 270–750 nm ($nm^{-1}$), slope from 275 to 295 nm, and slope 350 to 400 nm, all $nm^{-1}$. Slope ratio is the quotient of the 275–295 and 350–400 slopes.

| Source/salinity | Slope | Slope 275–295 | Slope 350–400 | Slope ratio |
| --- | --- | --- | --- | --- |
| Pre | | | | |
| 0 | 0.0141 | 0.0129 | 0.0170 | 0.762 |
| 10 | 0.0140 | 0.0131 | 0.0168 | 0.783 |
| 20 | 0.0137 | 0.0128 | 0.0164 | 0.782 |
| 35 | 0.0130 | 0.0121 | 0.0153 | 0.789 |
| Kirkpatrick | | | | |
| 0 | 0.0144 | 0.0129 | 0.0170 | 0.757 |
| 10 | 0.0142 | 0.0125 | 0.0167 | 0.746 |
| 20 | 0.0143 | 0.0125 | 0.0172 | 0.726 |
| 35 | 0.0139 | 0.0119 | 0.0168 | 0.707 |
| Taskinas | | | | |
| 0 | 0.0153 | 0.0135 | 0.0185 | 0.731 |
| 10 | 0.0156 | 0.0134 | 0.0191 | 0.703 |
| 20 | 0.0169 | 0.0161 | 0.0212 | 0.757 |
| 35 | 0.0142 | 0.0122 | 0.0171 | 0.716 |
| Jug Bay | | | | |
| 0 | 0.0156 | 0.0149 | 0.0189 | 0.789 |
| 10 | 0.0158 | 0.0149 | 0.0192 | 0.776 |
| 20 | 0.0171 | 0.0163 | 0.0216 | 0.756 |
| 35 | 0.0142 | 0.0128 | 0.0171 | 0.752 |
| Wachapreague | | | | |
| 0 | 0.0156 | 0.0143 | 0.0188 | 0.760 |
| 10 | 0.0155 | 0.0150 | 0.0185 | 0.810 |
| 20 | 0.0173 | 0.0157 | 0.0219 | 0.717 |
| 35 | 0.0142 | 0.0125 | 0.0169 | 0.741 |

cence components which were compared to published components using the online OpenFluor database (Murphy et al., 2014). The six components correspond to different types of CDOM found in aquatic environments with varying degrees of terrestrial influence (e.g., Stedmon and Markager, 2005; Lapierre and Del Giorgio, 2014; Pinsonneault et al., 2016) (Table 6). The first three components, C1 through C3, had spectral signatures resembling those of stable, humic compounds from terrestrial sources, while the spectral signatures of components C4 through C6 were protein-like resembling those of the comparatively more reactive amino acids phenylalanine, tryptophan, and tyrosine, respectively.

The adsorption–desorption incubations shifted the relative contribution of each of the three humic components to CDOM fluorescence by differential adsorption to the tidal marsh soils. This is shown by plotting "isotherms" of the initial minus final component scores ($\Delta C$, RU) vs. initial TDOC, with positive indicating net "adsorption" (loss to the solid phase) of that component (Fig. 7). Since C1 and C2 showed similar changes, we plotted the change in the sum of the two components (C1 + C2).

Net adsorption of component C3 was similar during incubation for all soils, salinities, and standards. The only exception was less net adsorption in the 0 treatment, especially for Taskinas. In contrast, the adsorption of C1 + C2 components varied widely with incubation salinity, with mainly net desorption for all $S = 0$ treatments of the brackish/saline marsh soils (Fig. 7a, b, and d). However, incubation in the strongly adsorbing soils of Jug Bay resulted in net adsorption of all humic components (Fig. 7c). Nevertheless, adsorption of C1 + C2 increased with salinity for all soils. This pattern of differential adsorption resulted in the net enrichment of C1 + C2 in the solute phase during incubation, with the most enrichment occurring at low salinity.

The protein-like components (C4 through C6) made minor contributions to EEM fluorescence with maximum scores for each component in the range of 0.21 to 0.86 RU compared to 1.50 to 3.86 RU for C1 through C3 (Table 6). The tryptophan-

**Table 6.** Parallel factor (PARAFAC) analysis component identification using a 0.97 similarity score as matching criteria in the OpenFluor database (Murphy et al., 2014).

| Component ID | $Ex_{max}$, $Em_{max}$ (nm) | Max score (RU) | Description | Literature source (component) |
|---|---|---|---|---|
| C1 | 275 (365), 498 | Pre: 2.07, post: 2.08 | UV–Vis humic-like, terrestrial, high MW and aromaticity, abundant in humic acid fractions extracted from sediments/soils. | Lambert et al. (2016) (C1) Walker et al. (2009) (C3) Yamashita et al. (2010) (C2) |
| C2 | 315, 422 | Pre: 1.59, post:1.50 | UV–Vis humic-like, terrestrial | Bittar et al. (2016) (C1) Yamashita et al. (2011) (C1) |
| C3 | < 245, 458 | Pre: 3.87, post: 2.06 | UV–Vis humic-like, terrestrial | Cawley et al. (2012) (C1) Dainard et al. (2015) (C1) Kothawala et al. (2014) (C5) |
| C4 | 245, 304 | Pre: 0.86, post: 0.07 | Protein-like, resembles spectra of phenylalanine | D'Andrilli et al. (2017) (C1) Jørgensen et al. (2011) (C3) Painter et al. (2018) (C1) |
| C5 | 285, 340 | Pre: 0.40, post: 0.45 | Protein-like, resembles spectra of tryptophan | Lambert et al. (2016) (C6) Murphy et al. (2008) (C7) Stedmon and Markager (2005) (C7) |
| C6 | 265, 288 (306) | Pre: 0.13, post: 0.21 | Protein-like, resembles spectra of tyrosine | D'Andrilli et al. (2017) (C2) Yamashita et al. (2013) (C3) |

like component (C5) content was proportional to the initial CDOC, ranging from 0 to 0.4 RU; otherwise there was no consistent relationship between the protein-like components and CDOC (data not shown). The phenylalanine-like component (C4) was prevalent in the standards with an average score of 0.30 RU but decreased during incubation to zero or near-zero. Post-incubation C4 scores only exceeded 0.02 RU in 12 of the 112 different incubation conditions, these being for low salinity in Taskinas, Jug Bay, and Wachapreague.

## 4 Discussion

Experiments presented here examined the interaction between highly colored, strongly humic, terrigenous DOC, and soils from tidal marshes of the Chesapeake Bay that vary in salinity regime and soil mineral content, among other factors. In an earlier analysis (Pinsonneault et al., 2021), we examined the changes in total dissolved organic carbon (TDOC) during soil-solution sorption incubations. Here we have further analyzed organic carbon interactions with tidal marsh soils by partitioning the changes in TDOC into colored (CDOC) and noncolored (NCDOC) fractions, tracking them in the dissolved phase.

Our analysis of the adsorption–desorption exchanges revealed that colored and noncolored organic matter have distinctly different adsorption–desorption characteristics that dramatically shifted the composition of DOC during incubations (Fig. 8). Noncolored matter was consistently desorbed in our experiments, regardless of the concentration of TDOC in the standards. The linear correlation of NC-DOC and CDOC in desorption isotherms shows that desorption of NCDOC is driven by the exchange (in most cases adsorption) of CDOC on soil surfaces. This coupled exchange was quantified as a displacement coefficient, which suggests that 12 %–44 % of CDOC adsorption is associated with NCOC desorption. Similarly, Kothawala et al. (2012) reported that incubating terrestrial mineral soils with aqueous extracts from organic horizons decreased the solution-specific absorbance and average molecular weights. They concluded that ligand exchange occurred on soil surfaces during the incubations whereby solution-phase aromatic C (strong ligands) replaced less aromatic (weaker ligands) soil C. This is consistent with our observation of strong adsorption of more aromatic CDOC and release into solution of the less aromatic NCDOC. Decrease in average molecular weight can also occur due to differential adsorption of CDOC components from the solution (cf. Fig. 7). However, this was a minor process under our experimental conditions because incubations had minimal effects on the slopes and slope ratios of absorbance spectra. Moreover, such differential adsorption would be expected to cause differential changes in the specific absorbance of CDOC (an indicator of average molecular weight) as a function of CDOC (cf. adsorption of fulvic acid on goethite Zhou et al., 2001), whereas we observed uniform specific absorbance over all incubation concentrations (Fig. 1).

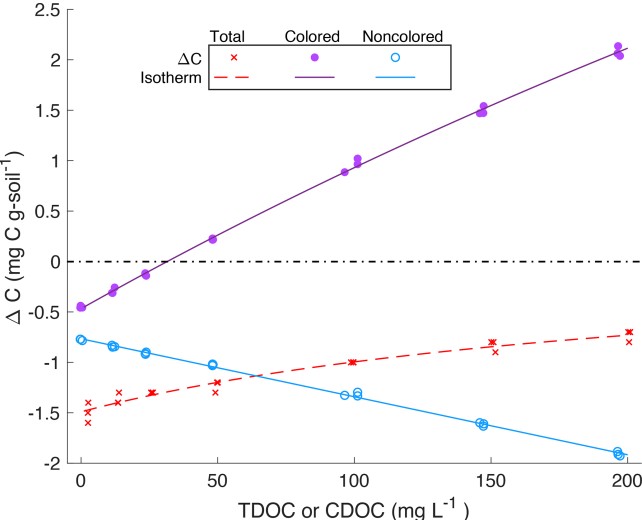

**Figure 4.** Comparison of sorption isotherms for incubation of DOC standards with soil from Kirkpatrick Marsh, $S = 10$, for TDOC (x, dashed line), CDOC (solid circles and line), and NCDOC (unfilled circles and solid line). Net adsorption of organic carbon, based on the difference between initial and final concentrations of soil carbon ($\Delta C$, mg C g-soil$^{-1}$), is plotted vs. solution TDOC (for total) or CDOC (colored and noncolored). Points show measured/estimated quantities from triplicate incubations for each standard; lines show the fitted Langmuir initial mass isotherm (TDOC, CDOC) and linear desorption isotherm (NCDOC). Dash–dot line indicates no net change; negative values indicate net desorption.

The specific absorbance of the CDOC derived from GDS surface waters was lower than that of Kirkpatrick low tide (LT) tidal surface water (Clark et al., 2019; see Appendix B, Fig. B3), so what we have designated as CDOC may be mixed with some NCDOC. This NCDOC could not be resolved by our approach because its concentration in the initial solution is simply a function of the dilution of the GDS stock. However, if there was NCDOC present in the incubation standards, it was not interacting with soil-adsorbed noncolored organic carbon. Any moiety in the standards that could exchange with the soil NCOC would act to repress net desorption of NCOC in incubations with sufficiently high TDOC. Such an interaction would result in a deviation from the observed tight linear relationship between NCDOC desorption and CDOC in the standards.

In contrast to NCDOC, the net adsorption or desorption of CDOC in the isotherm experiments followed the functional response of Langmuir isotherms, as observed for TDOC (Pinsonneault et al., 2021). However, $C_{C0}$ (native adsorbed COC) was much lower than $C_0$ for total organic carbon meaning that the transition from net desorption to net adsorption (null point) was at a much lower concentration for CDOC than for TDOC. This reflects the preferential adsorption of CDOC over most of the concentration range of the standards. Appreciable net desorption only occurred

in the highly organic soils from Kirkpatrick and Taskinas and at low $CDOC_i$ ($< 32$ mg L$^{-1}$ for saline soils, Fig. 3). Net adsorption of CDOC dominated in the Jug Bay and Wachapreague incubations. In contrast to $C_{C0}$, the isotherm parameters ($Q_{Cmax}$ and $K_C$) were well correlated for TDOC and CDOC, although the magnitudes were different (Fig. 6). The maximum adsorption capacity, $Q_{Cmax}$, was almost 4 times higher for CDOC than $Q_{Tmax}$ for TDOC, while $K_T$, the binding affinity, was about half. The magnitudes are different even though the adsorption of TDOC and CDOC follow similar patterns because the TDOC reflects the combination of CDOC adsorption and NCOC desorption. Nevertheless, the TDOC isotherm parameter correlations (Fig. 6) imply that the previously reported relationships between $Q_{Tmax}$ or $K_T$ and various soil properties also apply to CDOC isotherms (Pinsonneault et al., 2021). The tested soil properties were specific surface area (SSA), non-crystalline iron, and % organic content. Like for TDOC, $Q_{Cmax}$ increased and $K_C$ decreased with greater salinity, although the enhancement of maximum soil sorption capacity at higher salinity was somewhat less in soils enriched in poorly crystalline iron minerals. However, soil organic content, not its SSA, determined how much COC was available to be desorbed into low concentration standards. Such relationships between commonly measured soil properties and differential DOC sorption demonstrate the potential for spatial modeling of DOC functional group sorption across Chesapeake Bay coastal wetland soils.

Variations in incubation physiochemical conditions also affected DOM absorbance, but these variations do not substantially affect our interpretation of CDOC dynamics. Increasing salinity, and thus ionic strength, increases CDOM absorbance, with proportionally greater increases at longer wavelengths (Gao et al., 2015). This is attributed to deprotonation of DOM and results in a lowering of the spectral slope of CDOM absorption spectra and increase of specific absorbance, e.g., at 355 nm (Gao et al 2015). Consistent with these effects is the slight reduction of spectral slope with increasing salinity in both pre- and post-incubation spectra and the increase in specific absorbance (inverse of the decrease in DOC regression slope) of pre-incubation CDOC (Tables 1 and 6). However, increasing salinity, per se, does not account for the trend in the key change related to desorption of NCDOC during incubations, the decrease in specific absorbance (increase in DOC slope). This decrease was intensified by increasing salinity (Tables 1). Increasing pH also increases DOM-specific absorbance (Gao et al., 2015), and pH increased in all our incubations from $4.6 \pm 0.09$ in the standards to post-incubation values of $6.74 \pm 0.03$, $6.64 \pm 0.04$, $5.15 \pm 0.05$, and $6.94 \pm 0.10$ for Kirkpatrick, Taskinas, Jug Bay, and Wachapreague soils, respectively (Pinsonneault et al., 2021). Again, the increase in pH did not have an important effect on post-incubation absorbance since specific absorbance decreased during the incubation, the opposite direction from what would be expected from the pH change alone.

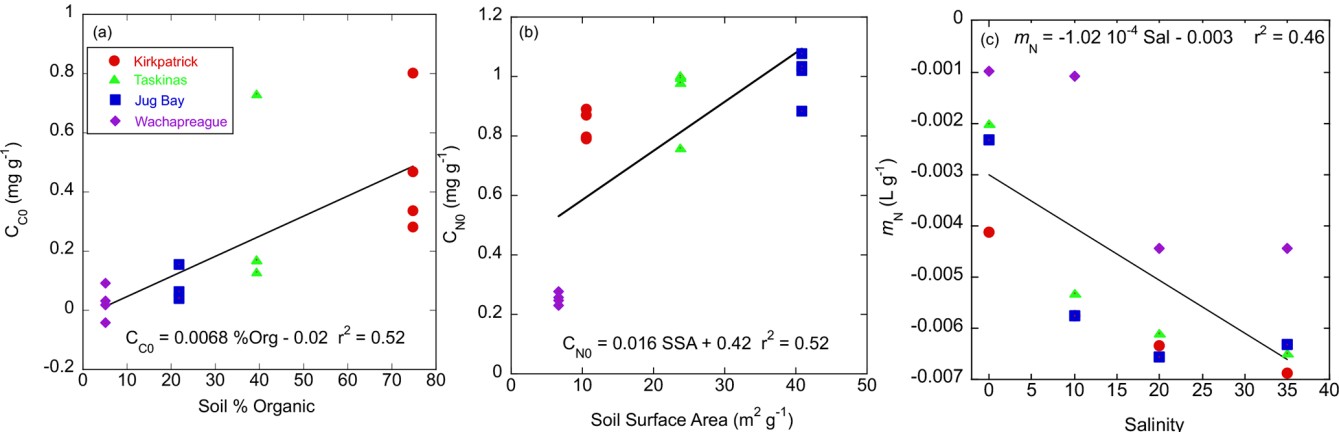

**Figure 5.** Relationship of sorption isotherm parameters for colored and noncolored organic carbon to soil characteristics. **(a)** $C_{C0}$ (mg g$^{-1}$) vs. % soil organic matter, **(b)** $C_{N0}$ (mg g$^{-1}$) vs. soil-specific area, **(c)** desorption coefficient ($m_N$, L g$^{-1}$) vs. incubation salinity. Red circles, green triangles, blue squares, and purple diamonds denote Kirkpatrick, Taskinas, Jug Bay, and Wachapreague, respectively.

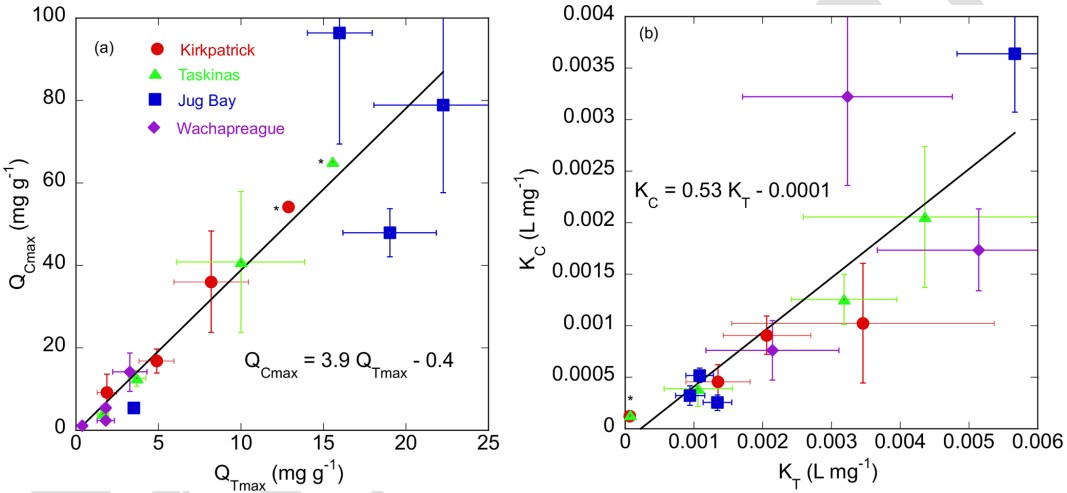

**Figure 6.** Comparison of maximum sorption capacity **(a)** and binding affinity **(b)** estimated for CDOC isotherms ($Q_{Cmax}$, $K_C$) vs. those for the TDOC isotherms ($Q_{Tmax}$, $K_T$) fit with initial mass Langmuir isotherms (the latter in Appendix B Table B2). Error bars indicate standard errors of the parameter estimates from non-linear regression. Red circles, green triangles, blue squares, and purple diamonds denote Kirkpatrick, Taskinas, Jug Bay, and Wachapreague, respectively. Points marked with asterisks are the Kirkpatrick and Taskinas 35 incubations, which were estimated by extrapolation and lack error bars.

Ferric (Fe$^{3+}$) iron also absorbs in the UV, and its concentration in some surface waters is high enough to significantly bias estimates of CDOM absorbance (Poulin et al., 2014; Logozzo et al., 2022). However, Fe is not a significant factor for DOM from the Great Dismal Swamp since it is a peat wetland with very low iron (U.S. Fish and Wildlife Service, 2006). Release of iron into solution during the incubation seems unlikely since it would have increased specific absorbance, again, the opposite of what we observed. Finally, we can exclude interference from dissolved iron on our absorbance measurements since all solutions were anaerobic, and dissolved iron would be in the non-absorbing Fe$^{2+}$ state.

In addition to releasing NCDOC, incubations with tidal marsh soils had secondary effects on the GDS CDOC pool. The humic C3 EEM component, which was the dominant component in the GDS pool, was strongly adsorbed during all incubation conditions (Fig. 7). In studies with a similar component (identified through OpenFluor, Table 6), C3 has been described as relatively resistant to photodegradation (Dainard et al., 2015), a property that may explain its relative abundance in the stagnant, sun-exposed ditch waters of the Great Dismal Swamp. On the other hand, adsorption of the humic components C1 and C2 was dependent on salinity, suggesting that these components account for the pattern of salinity-dependent adsorption observed in the CDOC (and

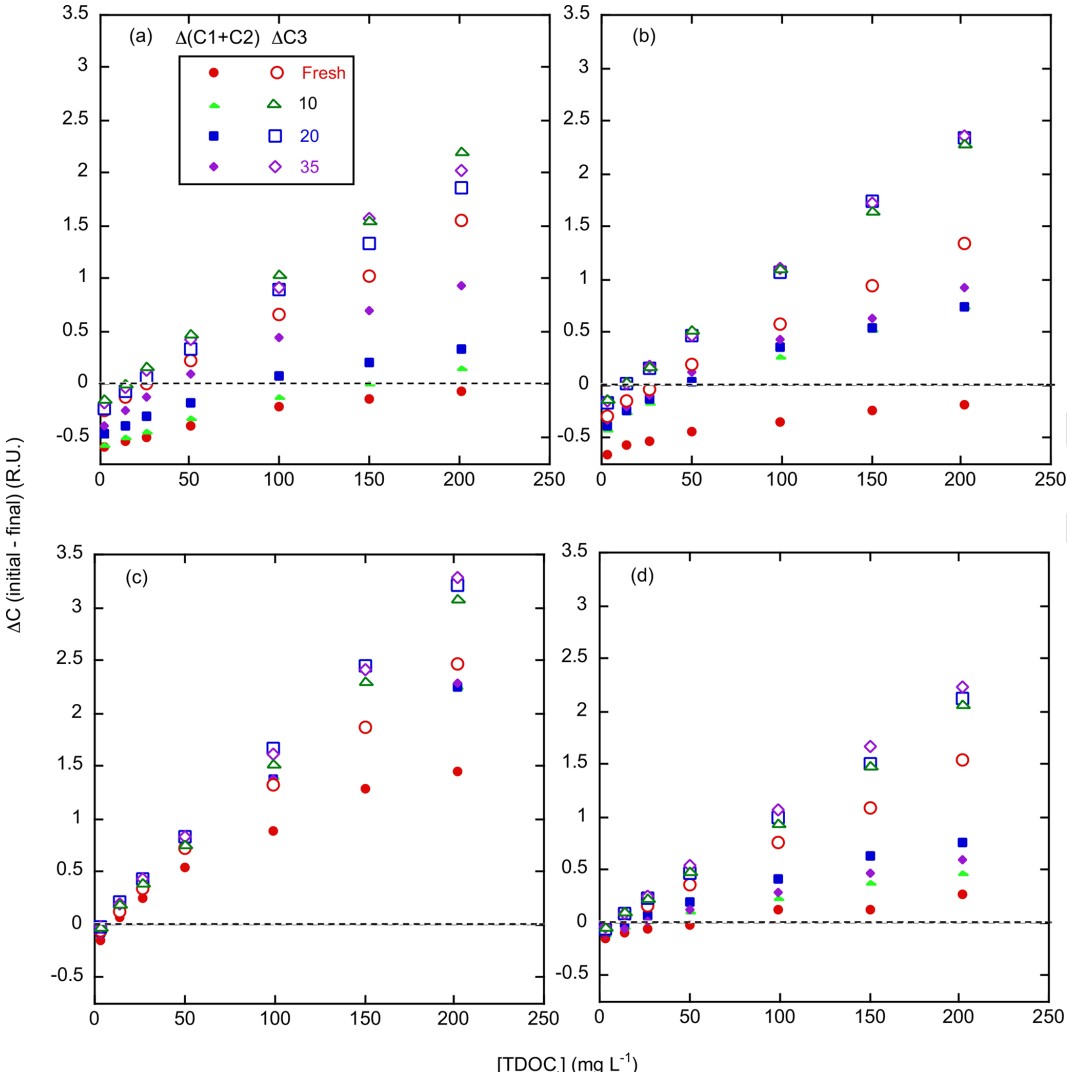

**Figure 7.** Change in EEM components C1–C3 during adsorption–desorption incubations. Plotted is the change in component score, initial – final (i.e., net "adsorption"), for the sum of C1 and C2 (RU, closed symbols) and C3 (RU, open symbols) as a function of initial incubation TDOC (TDOC$_i$, mg L$^{-1}$). **(a)** Kirkpatrick, **(b)** Taskinas, **(c)** Jug Bay, and **(d)** Wachapreague. Red circles, green triangles, blue squares, and purple diamonds reflect 0, 10, 20, and 35 salinity treatments, respectively.

TDOC) isotherms. The C4 component (phenylalanine-like) was also depleted from the dissolved fraction during incubation. Taking these results into account, we conclude that incubation with tidal marsh soils changes the composition of DOC primarily due to adsorption–desorption differences between noncolored and colored organic matter, but there can also be selective sorption that changes the composition of the colored organic matter pool. Further information on the selective effects of adsorption–desorption on the DOC composition of tidal marsh waters could be obtained by tracking the differential changes with high-resolution molecular techniques, such as FT-ICR-MS and high-resolution LC-MS. This information is essential for including tidal marsh sorp-

tion effects in biogeochemical models of carbon cycling in coupled tidal marsh–estuarine systems.

Overall, these results show that adsorption–desorption of TDOC and CDOC are similar, suggesting that sorption of CDOC accounted for most of the total DOC sorption in our marsh soils. Our experimental conditions – incubation of sieved, freeze-dried soil with non-native DOC – are not representative of adsorption–desorption in situ; nevertheless, the estimated CDOC null points for our soils are in the same range as porewater DOC measured at these sites. Spot samples of porewater using "peepers" (12 h deployment) had DOC concentrations between 10 and 40 mg C L$^{-1}$ (Pinsonneault et al., unpublished data), similar to the null points of CDOC in our IM isotherms (excluding the freshwater treat-

**Figure 8.** Conceptual diagram of the closed incubation experiments showing CE3 **(a)** pre- vs. **(b)** and **(c)** post-incubation exchange of colored (beige dots) vs. non-colored (white dots) organic carbon that is either sorbed to soil (bottom brown area) or dissolved (top blue-green area). The initial composition of the wetland soil sorbed organic carbon ($OC_{sorbed}$) is a mixture of colored and noncolored OC, whereas the Great Dismal Swamp DOC contains almost exclusively colored OC ($OC_{dissolved}$). During the incubation at low salinity **(b)**, colored OC is preferentially adsorbed to soil while non-colored OC is desorbed into solution. These net fluxes are intensified at high salinity **(c)**.

ments of two brackish sites, Kirkpatrick and Taskinas). This suggests that exchanges of colored organic matter adsorbed on these tidal marsh soils and CDOC in associated pore waters could be close to equilibrium. Such an equilibrium is consistent with the hypothesis that tidal marsh soils act as reservoirs of CDOM that buffer the release of CDOM into surface waters (Tzortziou et al., 2011).

In our experiments the noncolored organic matter fraction only desorbed, leaving open questions regarding its adsorption characteristics. Other types of test solutions may be needed to determine the adsorption–desorption characteristics of the noncolored fraction. Since test solutions from GDS were from static areas with long residence times, NCDOC comparable to NCOC in tidal soils may have been relatively low as the former fraction is presumably relatively more reactive to microbial degradation. On the other hand, water draining from Kirkpatrick tidal marsh at low tide, having been recently in contact with tidal marsh soil, has varying amounts (40 %–70 %) of its DOC as NCDOC (Clark et al., 2019); experiments with these solutions may help to better resolve the sorption properties of NCDOC.

### 4.1 Implications

While more needs to be known about the sorption properties of NCOC, our results lead us to predict that increasing salinity in tidal marsh ecosystems will shift the DOC composition of marsh porewater exported to estuaries, increasing the proportion of noncolored vs. colored organic matter (cf. Fig. 8). We predict that, all other factors held constant, shifts in the relative amounts of noncolored and colored organic matter will be greatest for mineral-rich soils, especially in soils with a very high surface area (SSA) like those of Jug Bay. The high SSA in our study soils arises, in part, from a high content of non-crystalline iron (Pinsonneault et al., 2021, Appendix B, Table B1). Coastal wetlands transitioning to a more saline condition may therefore export more bioreactive (i.e., less complex, noncolored) organic compounds, increasing substrate availability to estuarine microbes. At the same time, less release of colored organic matter in tidal waters would increase transmission of UV and PAR, increasing photochemical degradation – and potentially also microbial availability – of estuarine CDOC (Logozzo et al., 2020). These shifts, as well as interplay of multiple co-occurring processes, would need to be considered in modeling the effect of salinity change on the carbon cycle in coupled tidal marsh–estuarine systems (Clark et al., 2020), including in connection with the restoration of tidal exchange to artificially impounded coastal marshes (Kroeger et al., 2017; Wang et al., 2019; Eagle et al., 2022). In addition to sea level rise, disturbances like warming, eutrophication, and landscape development are all expected to alter the complex relationships between ecological, geochemical, and geomorphic controls on soil organic matter decomposition and preservation in coastal marsh systems (Spivak et al., 2019).

Chromophoric solutes have been used as indicators of water quality in a wide variety of ecosystems (e.g., Levy et al., 2014; Liu et al., 2014; Niu et al., 2014), and it has been reported that CDOM composition differs significantly between flooding and ebbing tides in tidal marshes (Tzortziou et al., 2008; Knobloch et al., 2022). Porewater draining from tidal creek banks at low tide is a primary source of CDOM in the ebbing tide (Menendez et al., 2022). Our results are consistent with this CDOM being derived by exchange with marsh sediment, as low DOC flood waters infiltrate into the sediment and desorb CDOC from the sediment. Our results also suggest that sediments are a significant source of NCDOC which, for example, comprises $\sim 50\%$ of the DOC in Kirkpatrick Marsh tidal creeks at low tide (Clark et al., 2019). However, our understanding of the abiotic processes that regulate this exchange and transformation at the tidal marsh–estuarine interface remains limited. Our study demonstrates the complex interplay between soil mineral and organic properties, salinity, and dissolved organic matter composition in regulating sorption processes at this dynamic interface, information that is required to represent the function and sensitivity of coastal interfaces in Earth system models (Ward

et al., 2020). Our results highlight the potential sensitivity of coastal wetlands to the impacts of anthropogenic climate change and evolving water management practices, resulting in heightened drought conditions and unpredictable variations in freshwater flow. We show that models forecasting salinity-driven changes to the fluxes and chemical characteristics of coastal wetland DOC export can succeed by incorporating soil parameters such as the poorly crystalline mineral content, soil organic matter content, and CDOC vs. NC-DOC composition as explicit variables. Further research into the impacts of small salinity increases ($< 10$) on the CDOC and NCDOC sorption dynamics in tidal freshwater and tidal oligohaline wetland soils would be instrumental in improving our understanding of the finer-scale and more immediate impacts of global climate change and/or management changes on sorption processes in these systems.

## Appendix A: Abbreviations

**Table A1.** List of abbreviations.

| | |
|---|---|
| $C_0$, $C_{C0}$, $C_{N0}$ | Initial natively adsorbed total, colored and non-colored organic carbon (respectively) (mg g$^{-1}$) |
| CDOC | Colored dissolved organic carbon (mg L$^{-1}$) |
| CDOM | Colored (or chromophoric) dissolved organic matter |
| COC | Colored organic carbon (e.g., sorbed to soil, mg g$^{-1}$) |
| DOM | Dissolved organic matter |
| $K_C$, $K_T$ | Binding affinity for CDOC and TDOC, respectively (L mg$^{-1}$) |
| $m_C$, $m_N$ | Linear desorption coefficient for COC and NCOC, respectively (L g$^{-1}$) |
| NCDOC | Non-colored dissolved organic carbon (mg L$^{-1}$) |
| NCOC | Non-colored organic carbon (e.g., sorbed to soil, mg g$^{-1}$) |
| $Q_{Cmax}$, $Q_{Tmax}$ | Soil maximum adsorption capacity for CDOC and TDOC, respectively (mg-C g$^{-1}$) |
| $S$ | Salinity on the practical salinity scale (unitless) |
| TDOC | Total dissolved organic carbon (colored + non-colored, mg L$^{-1}$) |
| TOC | Total organic carbon (%) |

## Appendix B:  Soil characteristics, TDOC isotherms, and dilutant absorbance

**Table B1.** Tidal marsh surface characteristics and soil characteristics for 0–40 cm depth including standard error. This table is presented as primary data in Pinsonneault et al. (2021), who present full descriptions of sample techniques and measurement methods. CE4

| Characteristic | Kirkpatrick Marsh | Taskinas Marsh | Jug Bay Marsh | Wachapreague Marsh |
|---|---|---|---|---|
| Soil core GPS coordinates | 38°52′25.90″ N 76°32′59.60″ W | 37°24′48.79″ N 76°42′57.44″ W | 38°46′52.32″ N 76°42′29.09″ W | 37°35′56.33″ N 75°38′9.38″ W |
| Dominant vegetation at soil | *Schoenoplectus americanus* | *Spartina patens Spartina* | *Typha latifolia Peltandra* | *Spartina alterniflora* |
| Core collection site | *Spartina patens Distichlis spicata* | *alterniflora Distichlis spicata* | *virginica Nuphar lutea* | *Salicornia europaea* |
| Surface water salinity | $7.15 \pm 0.55$ | $14.8 \pm 0$ | $1.29 \times 10^{-1} \pm 4.50 \times 10^{-3}$ | $32.0 \pm 0$ |
| % soil organic matter | $74.8 \pm 0.2$ | $39.3 \pm 1.8$ | $21.8 \pm 0.6$ | $5.12 \pm 0.54$ |
| % total organic carbon | $36.2 \pm 2.2$ | $12.4 \pm 0.3$ | $7.29 \pm 0.54$ | $1.77 \pm 0.12$ |
| % total nitrogen | $2.32 \pm 0.14$ | $0.763 \pm 0.021$ | $0.649 \pm 0.038$ | $0.130 \pm 0.008$ |
| TOC : TN | $15.6 \pm 0.3$ | $16.3 \pm 0.3$ | $11.1 \pm 0.3$ | $13.6 \pm 0.5$ |
| Bulk density ($\mathrm{g\,cm^{-3}}$) | $1.08 \times 10^{-1} \pm 8.11 \times 10^{-3}$ | $1.71 \times 10^{-1} \pm 1.25 \times 10^{-2}$ | $3.05 \times 10^{-1} \pm 3.42 \times 10^{-2}$ | $1.99 \pm 0.12$ |
| Specific surface area ($\mathrm{m^2\,g^{-1}}$) | $10.5 \pm 0.4$ | $23.8 \pm 1.7$ | $40.8 \pm 8.1$ | $6.60 \pm 1.09$ |
| Non-crystalline Al ($\mathrm{mg\,g^{-1}}$) | $5.92 \times 10^{-1} \pm 3.02 \times 10^{-2}$ | $1.43 \pm 0.08$ | $1.22 \pm 0.01$ | $4.64 \times 10^{-1} \pm 1.33 \times 10^{-2}$ |
| Non-crystalline Fe ($\mathrm{mg\,g^{-1}}$) | $1.37 \times 10^{-1} \pm 3.54 \times 10^{-2}$ | $1.39 \pm 0.46$ | $11.2 \pm 1.7$ | $1.30 \pm 0.28$ |
| % sand | $31.6 \pm 9.8$ | $46.2 \pm 8.4$ | $31.9 \pm 6.6$ | $40.6 \pm 2.3$ |
| % silt | $60.4 \pm 7.6$ | $40.8 \pm 6.1$ | $57.0 \pm 5.3$ | $50.5 \pm 2.3$ |
| % clay | $8.07 \pm 2.4$ | $13.0 \pm 2.3$ | $11.1 \pm 1.3$ | $8.95 \pm 0.13$ |
| Mean grain size (μm) | $39.8 \pm 11.9$ | $66.6 \pm 30.6$ | $35.2 \pm 8.7$ | $49.1 \pm 3.2$ |

**Table B2.** Mean and standard error of sorption characteristics for total DOC of tidal marsh soils based on fits to the initial mass (IM) Langmuir isotherm by salinity treatment (see Fig. B2). $Q_{\mathrm{Tmax}}$, $C_0$, $K_{\mathrm{T}}$, and NP refer to the soil maximum sorption capacity, initial exchangeable soil carbon, the DOC binding affinity, and the null point, respectively. Data for the fit were presented in Pinsonneault et al. (2021), where they were fit to the traditional Langmuir isotherm and here were refit to the IM Langmuir isotherm. Values in parentheses were estimated by extrapolation from other salinities; these were not used to estimate NP. CE5

| Variable | Kirkpatrick | Taskinas | Jug Bay | Wachapreague |
|---|---|---|---|---|
| $Q_{\mathrm{Tmax}}$ ($\mathrm{mg\,g^{-1}}$) | | | | |
| 0 | $1.9 \pm 0.6$ | $1.6 \pm 0.4$ | $3.5 \pm 0.2$ | $0.4 \pm 0.0$ |
| 10 | $4.9 \pm 1.1$ | $3.7 \pm 0.5$ | $19.0 \pm 2.8$ | $1.8 \pm 0.5$ |
| 20 | $8.2 \pm 2.3$ | $10.0 \pm 3.9$ | $16.0 \pm 2.0$ | $1.8 \pm 0.3$ |
| 35 | $(12.9)$ | $(15.5)$ | $22.3 \pm 4.2$ | $3.3 \pm 1.0$ |
| $C_0$ ($\mathrm{mg\,g^{-1}}$) | | | | |
| 0 | $1.60 \pm 0.04$ | $1.49 \pm 0.03$ | $1.04 \pm 0.03$ | $0.34 \pm 0.03$ |
| 10 | $1.26 \pm 0.02$ | $1.17 \pm 0.03$ | $1.12 \pm 0.02$ | $0.22 \pm 0.03$ |
| 20 | $1.23 \pm 0.02$ | $1.15 \pm 0.03$ | $1.10 \pm 0.02$ | $0.30 \pm 0.03$ |
| 35 | $1.15 \pm 0.01$ | $1.13 \pm 0.02$ | $1.06 \pm 0.02$ | $0.26 \pm 0.02$ |
| $K_{\mathrm{T}}$ ($\mathrm{L\,mg^{-1}}$) $\times 10^{-3}$ | | | | |
| 0 | $3.46 \pm 1.91$ | $4.36 \pm 1.77$ | $5.67 \pm 0.85$ | $22.27 \pm 9.19$ |
| 10 | $2.06 \pm 0.64$ | $3.18 \pm 0.77$ | $1.09 \pm 0.20$ | $3.23 \pm 1.52$ |
| 20 | $1.35 \pm 0.47$ | $1.06 \pm 0.50$ | $1.34 \pm 0.21$ | $5.14 \pm 1.47$ |
| 35 | $(0.07)$ | $(0.08)$ | $0.95 \pm 0.21$ | $2.14 \pm 0.96$ |
| NP ($\mathrm{mg\,L^{-1}}$) | | | | |
| 0 | 1668 | 2458 | 274 | 74 |
| 10 | 169 | 147 | 42 | 58 |
| 20 | 131 | 123 | 38 | 55 |
| 35 | n/a TS13 | n/a | 40 | 53 |
| $r^2$ | | | | |
| 0 | 0.918 | 0.958 | 0.988 | 0.893 |
| 10 | 0.988 | 0.982 | 0.995 | 0.975 |
| 20 | 0.990 | 0.985 | 0.996 | 0.980 |
| 35 | 0.994 | 0.981 | 0.992 | 0.983 |

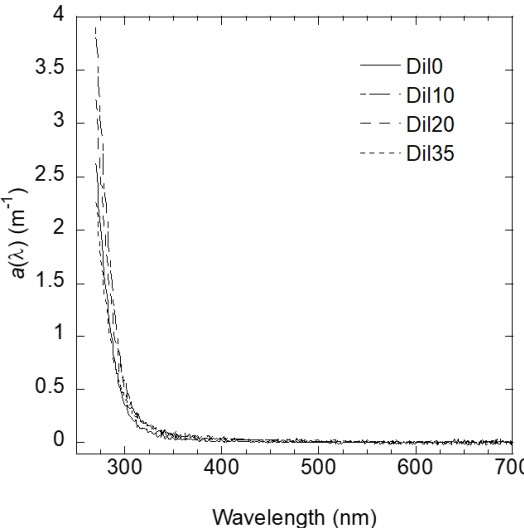

**Figure B1.** Average absorbance spectra ($a(\lambda)$, m$^{-1}$) of dilutant used to create different standards from the Great Dismal Swamp concentrate. Line type indicates dilutant for fresh treatment (solid line), 10 (long dashes), 20 (medium-length dashes), and 35 salinity treatments (short dashes).

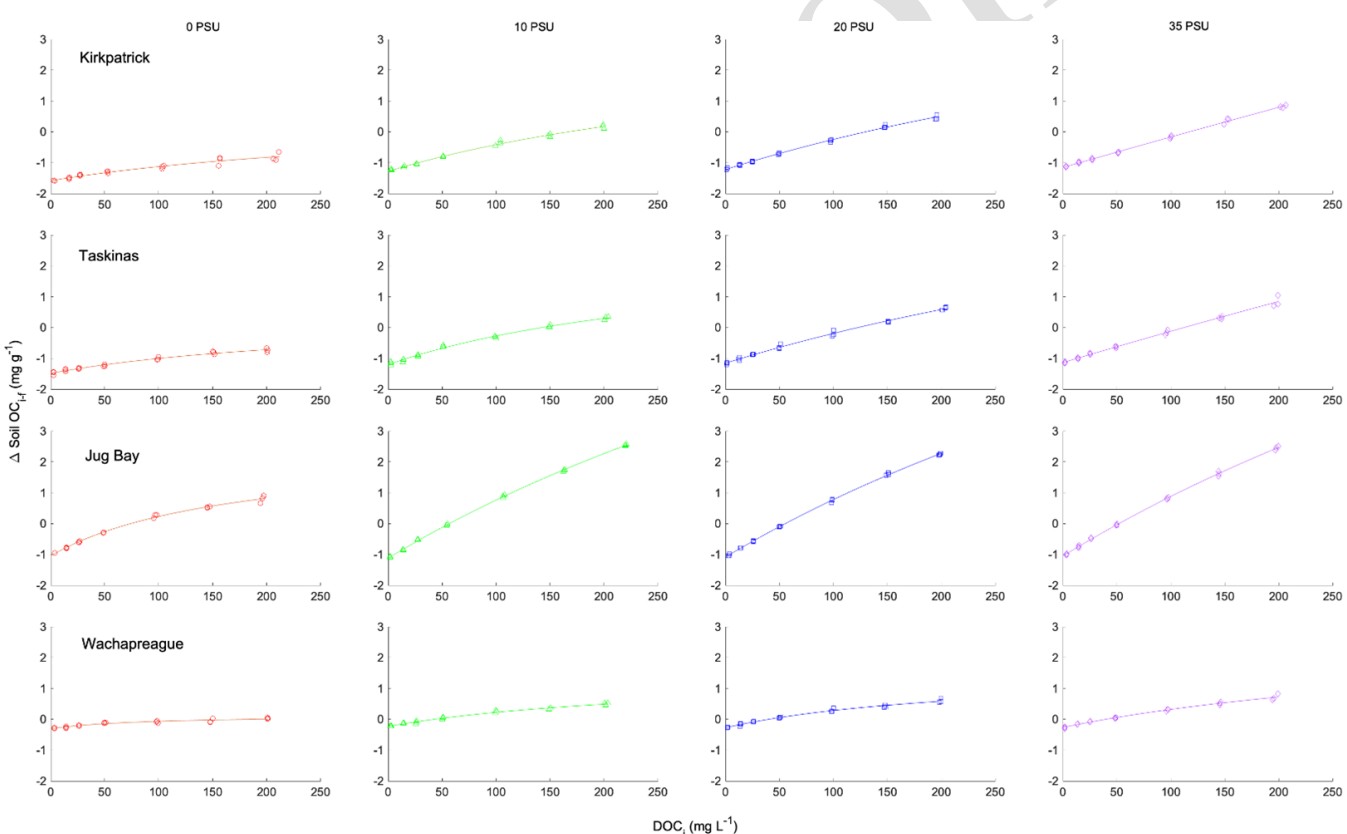

**Figure B2.** Isotherms of adsorption–desorption of TDOC for tidal marsh soils. Net adsorption of organic carbon based on the difference between initial and final concentrations of soil organic carbon ($\Delta$C, mg C g-soil$^{-1}$) as a function of initial concentration of DOC ([DOC]$_i$, mg L$^{-1}$) using data from Pinsonneault et al. (2021). Symbols show the results of triplicate incubations, lines the fitted IM isotherm. By rows the plots show (from top to bottom) Kirkpatrick, Taskinas, Jug Bay, and Wachapreague. Red circles, green triangles, blue squares, and purple diamonds reflect 0, 10, 20, and 35 salinity treatments, respectively.

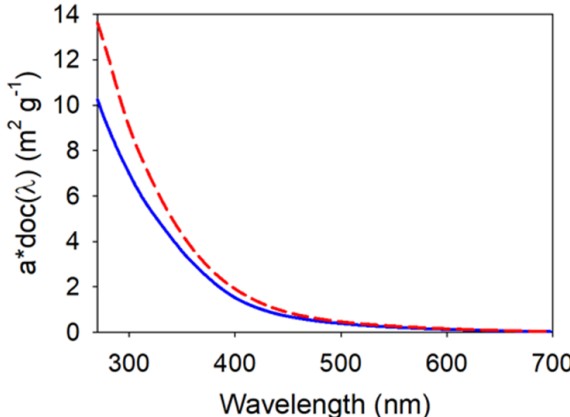

**Figure B3.** Plot of the carbon-specific absorption spectra ($a \cdot$ TS14 CDOC($\lambda$), m$^2$ g$^{-1}$) of Great Dismal Swamp standards (this study, solid blue line) and Kirkpatrick Marsh low tide water (Clark et al., 2019, red dashed line).

*Data availability.* All data used for the publication can be downloaded from Zenodo: https://doi.org/10.5281/zenodo.10845926 (Patrick, 2024 TS15 ).

*Supplement.* The supplement related to this article is available online at: https://doi.org/10.5194/bg-21-1-2024-supplement.

*Author contributions.* All authors contributed to the study conception and design. PJN and JPM supervised the data collection, and the data analysis was performed by PJN and HM. MT, JPM, PJN, and EAC were responsible for project administration and funding acquisition. PJN led the manuscript writing, and all authors revised drafts and approved the final manuscript.

*Competing interests.* The contact author has declared that none of the authors has any competing interests.

ther geographical representation in this paper. While Copernicus Publications makes every effort to include appropriate place names, the final responsibility lies with the authors.

*Acknowledgements.* We would like to thank the staff of the Jug Bay Wetlands Sanctuary, the York River State Park, the Virginia Institute of Marine Science (VIMS) Eastern Shore Laboratory, and the Great Dismal Swamp National Wildlife Refuge, for their support and assistance. In particular, we are grateful to Amanda Knobloch, Yanhua Feng, and Sean Fate (VIMS), Willy Reay (Chesapeake Bay National Estuarine Research Reserve), Michael Gonsior (University of Maryland Center for Environmental Science, Chesapeake Biological Laboratory), Jade Dominique Walker (University of Maryland University College), Ellen Weber (Wilkes University), and Andrew Peresta and Andrew Pinsonneault (Smithsonian Environmental Research Center) for their assistance in the field and laboratory. This study was funded by the National Science Foundation (NSF) grant nos. DEB-1556556 (Maria Tzortziou, J. Patrick Megonigal and Patrick J. Neale) and DEB 1556554 (EAC) and NSF Long-Term Research in Environmental Biology Program grant nos. DEB-0950080, DEB-1457100, DEB-1557009, and DEB-2051343 (J. Patrick Megonigal ). The work was supported by the Smithsonian Institution, the Smithsonian Environmental Research Center, and the Global Change Research Wetland. Funding was also provided by the US Department of Energy, Office of Science, Office of Biological and Environmental Research, Environmental System Science program through the COMPASS-FME project (grant no. DE-AC05-76RL01830).

*Financial support.* TS16 This research has been supported by the National Science Foundation (grant nos. DEB-1556556, 1556554, 0950080, 1457100, 1557009, 2051343) and the Biological and Environmental Research (grant no. DE-AC05-76RL01830).

*Review statement.* This paper was edited by Ji-Hyung Park and reviewed by two anonymous referees.

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

**Remarks from the language copy-editor**

CE1    Should these be normal parentheses? Please note that we use square brackets to denote concentrations. Please double-check this throughout.

CE2    Please double-check the content of this table, as adjustments unfortunately do not show up in the track-changes PDF.

CE3    The descriptions for a, b, and c need to be reworded.

CE4    Please double-check the content of this table, as adjustments unfortunately do not show up in the track-changes PDF.

CE5    Please double-check the content of this table, as adjustments unfortunately do not show up in the track-changes PDF.

**Remarks from the typesetter**

TS1    Please provide department if possible.

TS2    Please provide department if possible.

TS3    Please provide reference list entry.

TS4    Please check. Is this a variable or parameter?

TS5    Please provide reference list entry.

TS6    Please provide reference list entry.

TS7    Please confirm change to capital letter "C" here and throughout the text.

TS8    Do you mean "not available" or "not applicable"? Please define.

TS9    Please confirm dot sign.

TS10    Please note: the asterisk is not mentioned in the table below.

TS11    Please confirm dot sign.

TS12    Please provide reference list entry.

TS13    Do you mean "not available" or "not applicable"? Please define.

TS14    Please confirm dot sign.

TS15    Please confirm addition.

TS16    Please note that the funding information has been added to this paper. Please check if it is correct. Please also double-check your acknowledgements to see whether repeated information can be removed or changed accordingly. Thanks.

TS17    Please ensure that any data sets and software codes used in this work are properly cited in the text and included in this reference list. Thereby, please keep our reference style in mind, including creators, titles, publisher/repository, persistent identifier, and publication year. Regarding the publisher/repository, please add "[data set]" or "[code]" to the entry (e.g., Zenodo [code]).

TS18    Please provide volume.

TS19    Please provide publisher and DOI or ISBN.

TS20    Please confirm addition.

TS21    Please provide publisher and DOI or ISBN.

TS22    Please provide date of last access (day month year).