# Peer review of "Sorption of Colored vs Noncolored Organic Matter by Tidal Marsh Soils"

_EGUsphere, 2023_

## Author Comment (AC1)

**Main Comments:**

**RW1:** The one major nagging question that I had throughout the entire review was that the authors have not accounted for, or even discussed anywhere, the potential limitations of using absorbance alone to partition colored versus non-colored DOC concentrations. Specifically, can shifts in absorbance in response to salinization induce optical changes in the DOM pool that alter absorbance through concentration-independent processes? Salt additions that alter ion binding patterns in the soil matrix could liberate/precipitate stuff like iron that may impact absorbance, no? It could also alter the chelation of DOM and metals and other materials, altering optical conditions? I am not raising these potential mechanisms to say that they invalidate the results. It is likely that at these DOC concentrations, such effects would be less important. BUT, using knowledge from the literature, and any other information on hand, can the authors speak to what role, if any, these effects have on their interpretations and conclusions (even if only to caveat them)?

**Author:** The reviewer raises two good points, one that solution salinity affects the optical properties of CDOM and secondly that salinity affects the binding to the soil matrix of other components, like iron, that may also affect absorbance properties. The first effect can be seen from the results in Table 6 which show slight decreases in CDOM slopes and slope ratios within each sample category as salinity is increased. Salinity increases are also linked to increased specific- absorbance at long wavelengths (Gao et al. 2015), but this effect contrasts with the observed trend in our post-incubation data of decreased specific absorbance (increased DOC slope, Table 1). As far as possible metal:DOC interactions in the GDS solution, the DOM was sampled from a freshwater peat marsh with very low soil iron – which is why DOM is high to begin with. Moreover, all absorbance measurements were made under anaerobic conditions. Thus dissolved iron, to the extent it was present, would mainly be in non-absorbing $Fe^{2+}$ form not the UV absorbing $Fe^{3+}$ form. We now include new text describing the absorbance methods (this text was inadvertently left out of our first submission) which notes that scans were done under anaerobic conditions:

*"Due to the highly colored nature of the Great Dismal Swamp DOC, we diluted subsamples of pre-incubation standards by a factor of 10 using dilutant from the same salinity treatment while we diluted the filtered post-incubation supernatant by the same factor using ultrapure water + NaCl (Sigma Aldritch, 99.5% purity) to match sample salinity. All solutions were degassed with $N_2$, then dilutions were performed, cuvettes loaded and sealed in an anaerobic chamber. We performed absorbance scans at 2 nm intervals (270 – 750 nm) for all replicates using a Thermo Scientific Evolution 220 UV-Vis spectrophotometer."*

In addition, we will add comments on the effects of salinity and iron as well as pH change (a request from another reviewer) on our absorbance results to the discussion (following the paragraph discussing how isotherm parameters varied with salinity):

*"We have considered possible physicochemical effects on DOM absorbance in our incubations, but these do not appear to substantially affect our interpretation of CDOC dynamics. Increasing*

*salinity, and thus ionic-strength, increases CDOM absorbance, with proportionally greater increases at longer wavelengths (Gao et al., 2015). This is attributed to deprotonation of DOM, and results in a lowering of the spectral slope of CDOM absorption spectra and increase of specific absorbance, e.g. at 355 nm (Gao et al 2015). While there was some increase in spectral slope and specific-absorbance with increasing treatment salinity for the pre-incubation standards (see Tables 1 and 6), we observed the opposite effect in the post-incubation solutions, i.e. adding salinity increased DOC vs absorbance slope (Table 1) which implies decreased specific absorbance. Increasing pH also increases DOM absorbance (Gao et al., 2015), and postincubation pH increased in all our incubations from 4.6±0.09 in the standards to 6.74±0.03, 6.64±0.04, 5.15±0.05 and 6.94±0.10 for Kirkpatrick, Taskinas, Jug Bay and Wachapreague soils (respectively) (Pinsonneault et al., 2021). The increase in pH did not have an important effect on postincubation absorbance since specific absorbance decreased during the incubation, the opposite direction from what would be expected from the pH change alone.*

*Ferric ($Fe^{3+}$) iron also absorbs in the UV and its concentration in some surface waters is high enough to significantly bias estimates of CDOM absorbance (Poulin et al., 2014; Logozzo et al., 2022). However, Fe is not a significant factor for DOM from the Great Dismal Swamp since it is a peat wetland with very low iron (USFWS, 2006). Release of iron into solution during the incubation seems unlikely since it would have increased specific absorbance, again, the opposite of what we observed. Finally, we can exclude interference from dissolved iron on our absorbance measurements since all solutions were anaerobic and dissolved iron would be in the non-absorbing $Fe^{2+}$ state."*

**Specific Comments:**

**RW1:** L23- GDS DOC – mention what this is at the start of abst. It's introduced in a strange spot.

**Author:** The source of the test solution will be moved to the first sentence describing our experiments:

*"To test this hypothesis, we generated initial mass sorption isotherms of CDOC and noncolored dissolved organic carbon (NCDOC) using anaerobic batch incubations of Great Dismal Swamp DOC with four tidal wetland soils, representing a range of organic carbon content (1.77 ± 0.12 % to 36.2 ± 2.2%) and across four salinity treatments (0, 10, 20, and 35)."*

**RW1:** L33 - Intro – Polydisp. – define what this is in brackets.

**Author:** Definition will be added to the sentence (underlined):

*"Studies have found that CDOM composition in flooding and ebbing tidal waters differs with more strongly colored and aromatic CDOM of higher average molecular weight and polydispersity [breadth of the molecular weight distribution] being exported from tidal marshes into estuarine waters during ebbing tide relative to that imported into marshes during flooding tide (Tzortziou et al., 2008; Tzortziou et al., 2011)."*

**RW1:** L58 – Run on sentence.

**Author:** Not clear to which sentence the reviewer is referring. We have checked the text in this part of the mss to ensure that we don't have a run on sentence.

**RW1:** L96 – What is a "filter cale"? Can you reword with something more generic? I have never seen the word "cale" before like this.

**Author:** This was a typo, will be corrected to "filter capsule"

**RW1:** L106- Is the NaN3 a preservative? Explain purpose briefly.

**Author:** Description will be added to sentence (underlined):

*"The resulting filtered DOC concentrate was then treated with 1 mM sodium azide (NaN$_3$), a microbial inhibitor, and DOC concentration ([DOC]) was measured with a Shimadzu TOC-L using high-temperature combustion."*

**RW1:** L123- Any details about this regression R2/strength criteria used? Were relationships scrutinized in any way?

**Author:** Regression $r^2$ values and parameter standard errors are presented in Table 2, the $r^2$ values were all very near 1.

**RW1:** L163- Fig 1 caption. Explain blue solid/dashed arrows so readers don't need to sift through text to interpret figure.

**Author:** Arrows will be identified in the caption:

*"Figure 1. Scatterplot of DOC content vs absorption coefficient at 355 nm in pre-incubation (triangles) and post-incubation (+ symbols) treatments in triplicate from the Kirkpatrick soil, S=10, isotherm experiment with fitted linear regressions. The dashed arrows indicate the difference in DOC between the regression lines for selected post-incubations absorbances, the solid arrows identify the DOC of the pre-incubation line at these absorbances."*

**RW1:** L204 – Fig 3. I spent a bunch of time trying to figure this out, then realized you are using negative values to mean desorption. Please state this up front in the caption. Even better would be to add a dashed zero line to signify the difference.

**Author:** The figures did have a zero gray line, this will be made into a dashed line that will be more visible. Also, we will note the meaning of negative values in the figure caption:

*"Figure 3. Isotherms of adsorption-desorption of CDOC (filled shapes) and NCDOC (open shapes) for tidal marsh soils. Net adsorption of organic carbon based on the difference between initial and final concentrations of soil organic carbon (ΔC, mg C g-soil-1) as a function of initial concentration of CDOC ([CDOC]i, mg L-1). Negative values indicate net desorption. Error bars show standard deviation of triplicate incubations, in most cases error bars are smaller than the symbol. Incubations used soils from (a) Kirkpatrick, (b) Taskinas, (c) Jug Bay, and (d) Wachapreague marshes. Red circles, green triangles, blue squares, and purple diamonds denote salinities of 0, 10, 20, and 35, respectively."*

**RW1:** L215- All of this isotherm theory and mathematical model interpretation would be far easier to follow as a section fully developed in the methods, perhaps with a conceptual diagram of the isotherm graph.

**Author:**  The text describing the linear desorption isotherm will be moved to the methods. Figure 4 was included to assist the reader in visualizing the context for the different isotherm models (see comment below).

**RW1:** L240- Throughout results, the ability for readers to wade through this info would be improved with the use of sub-headings to group info by themes.

**Author:** We will add additional sub-headings in the results,  3.2 CDOC Sorption Isotherms, 3.3 NCDOC Desorption Isotherms and 3.4 Relationship of Isotherm Parameters to Soil Characteristics

**RW1:** L251- Figure 4- Confusing. Is the y axis still initial minus final? If so, make that one clear.

**Author:** This is stated clearly in the figure caption*.*

*"Net adsorption of organic carbon, based on the difference between initial and final concentrations of soil carbon (ΔC, mg C g-soil-1), is plotted vs solution TDOC (for Total) or CDOC (Colored and Noncolored)."*
To further emphasize the occurrence of net desorption for some initial concentrations of CDOC a dashed line will be added at zero.

**RW1:** L273- Again, a bunch of this background definition content would be nice to see in a dedicated methods section up front.

**Author:**  The definition of null point and description of its meaning will be moved to methods:

*"The null point (NP) is defined as the concentration at which there is no net removal of CDOC from solution (adsorption) or release from soil (desorption), the biogeochemical significance of which is a sorptive equilibrium between the soil mineral surfaces and the aqueous phase.  It is derived by setting Eq 2 to zero and solving for NP concentration of [CDOC]i:*

$$NP = \frac{C_{C0}}{(Q_{Cmax}-C_{C0})*K_c}$$ (3)"

**RW1:** L281- Initial sentence not needed.

Author:  We agree that it was not needed for this paragraph.  The purpose was to provide a segue to the spectral results, which is better accomplished by having a rephrased version as the topic sentence of the following paragraph:

"*Given that spectral characteristics of CDOM absorbance and fluorescence are used to infer sources, sinks, reactivity, and other biogeochemical processes (Chin et al., 2002; Helms et al., 2008; Tzortziou et al., 2008), we examined how spectral properties of CDOC were affected by sorption processes.*"

**RW1:** L293- "than less" – grammar.

**Author:** Changed to "and less"

**RW1:** L330- Do the authors find this odd that these components decrease? NCDOC increases, so should these components not also go up? I am not saying that they must, I am likely missing something here, but just want the authors to explain this to me/readers.

**Author:**  It is not clear to which component decrease the reviewer is referring.  Line 333 speaks of the decrease in the phenylalanine component, C4.  As something prevalent in the GDS, it is regarded as a component of the CDOC, not the NCDOC, so there is no inconsistency with the increase in NCDOC.

**RW1:** L345- This figure feels a bit unnecessary/of limited use in current format. I'd swap out the text "dismalswamp" with something generic so that someone outside the region will understand is the stock solution without knowing the location it came from, and not having to consult the methods again. The grey/brown DOC could be defined in a legend so it is visually obvious what the color coding means.  The figure could be redone to reflect what increasing salinity does to DOC sorption, which is a key question as I take it in the paper.

**Author:** Based on the reviewer's helpful comments the figure will be revised:

[Figure]

[Figure]

The different phases are now described in general terms and the color code for CDOC and NCDOC is explicit in the diagram.  Separate panels will be shown to depict the increase in net adsorption of CDOC and desorption of NCDOC at high salinity.

**RW1:** L362- Ref format in brackets, run on sentence as written.

**Author:**  The text will be broken up into two sentences:

*"However, this was a minor process under our experimental conditions because incubations had minimal effects on the slopes and slope ratios of absorbance spectra.   Moreover, such differential adsorption would be expected to cause differential changes in the specific absorbance of CDOC (an indicator of average molecular weight) as a function of CDOC (cf. adsorption of fulvic acid on goethite Zhou et al., 2001) whereas we observed uniform specific absorbance over all incubation conditions (Fig. 1)."*

**RW1:** L364- Why redefining GDS here?

**Author:**  Abbreviation redefinition deleted.

**RW1:** L393- Photodegradation reference needed here.

**Author:**  Citation added to Dainard et al (2015).

**RW1:** L420- If the conceptual figure could link to this sentence it would be more useful. See my comment above about incorporating salinity.

**Author:**  The figure is now modified to illustrate our interpretation of the salinity effect

**RW1:** L437- You come back to flood/ebb differences in DOM here. A few more sentences integrating the mechanisms exposed by the experiment, and how they potentially help to explain the empirical observations in the literature, would strengthen the discussion.

**Author:**  We will add new text placing sorption processes in the context of flood/ebb differences:

*"Porewater draining from tidal creek banks at low-tide is a primary source of CDOM in the ebbing tide (Menendez et al., 2022). Our results are consistent with this CDOM being derived by exchange with marsh sediment, as low DOC flood waters infiltrate into the sediment and desorb CDOC from the sediment.  Our results also suggest that sediments are a significant source of NCDOC which, for example, comprises ~50% of the DOC in Kirkpatrick Marsh tidal creeks at low tide"*

References:

Gao, Y., Yan, M., and Korshin, G. V.: Effects of Ionic Strength on the Chromophores of Dissolved Organic Matter, Environ. Sci. Technol., 49, 5905-5912, 10.1021/acs.est.5b00601, 2015.

Logozzo, L. A., Martin, J. W., McArthur, J., and Raymond, P. A.: Contributions of Fe(III) to UV–Vis absorbance in river water: a case study on the Connecticut River and argument for the systematic tandem measurement of Fe(III) and CDOM, Biogeochemistry, 160, 17-33, 10.1007/s10533-022-00937-5, 2022.

Menendez, A., Tzortziou, M., Neale, P., Megonigal, P., Powers, L., Schmitt-Kopplin, P., and Gonsior, M.: Strong Dynamics in Tidal Marsh DOC Export in Response to Natural Cycles and Episodic Events From Continuous Monitoring, J. Geophys. Res.-Biogeo., 127, 10.1029/2022jg006863, 2022.

Poulin, B. A., Ryan, J. N., and Aiken, G. R.: Effects of Iron on Optical Properties of Dissolved Organic Matter, Environ. Sci. Technol., 48, 10098-10106, 10.1021/es502670r, 2014.

U.S. Fish and Wildlife Service: Great Dismal Swamp and Nansemond NWRs Comprehensive Conservation Plan, https://www.fws.gov/media/great-dismal-swamp-and-nansemond-nwrs-comprehensive-conservation-plan, 2006.

Citation: https://doi.org/10.5194/egusphere-2023-2329-RC1

---

## Author Comment (AC2)

**Main Comments:**

**RW2:** The methods lack information on precision and trueness of TOC, DOC and metals measurements. EEMs are introduced in the results only. I understand the reasoning, but it still comes as a bit of a surprise and lacks important information on measurements. This should be more detailed and go into the methods section. In my view the rationale can be mentioned without giving away too much of the results.

**Author:** We inadvertently omitted the methods description for absorbance and fluorescence which will now be provided:

"*Due to the highly colored nature of the Great Dismal Swamp DOC, we diluted subsamples of pre-incubation standards by a factor of 10 using dilutant from the same salinity treatment while we diluted the filtered post-incubation supernatant by the same factor using ultrapure water + NaCl (Sigma Aldritch, 99.5% purity) to match sample salinity. All solutions were degassed with $N_2$ and samples handled in an anaerobic chamber. We performed absorbance scans at 2 nm intervals (270 – 750 nm) for all replicates using a Thermo Scientific Evolution 220 UV-Vis spectrophotometer. Specific ultraviolet absorbance at 280 nm ($SUVA_{280}$), an indicator of DOC aromaticity, was calculated from this data by dividing decadic sample absorbance at 280 nm by the DOC concentration ([DOC]) (Hansen et al., 2016). We then generated three-dimensional excitation-emission matrices (EEMs) using a Horiba Jobin Yvon FluoroMax-3 spectrofluorometer (for sample replicate A only) at 5 nm intervals (250 – 600 nm) for excitation and 2 nm intervals (250 – 600 nm) for emission. Fluorescence spectra were corrected for inner-filter effect and Raman scattering using the drEEM toolbox version 0.2.0 (Murphy et al. 2013) in MATLAB (v. 2017b). Parallel factor analysis (PARAFAC) was used to deconstruct the fluorescence signal into underlying fluorescence components, or fluorophores, that relate to differences in DOC composition (Murphy et al., 2010; Lapierre and del Giorgio, 2014).*"

Metal measurements are reprinted in Appendix Table B1 from Pinsonneault et al. (2021). Details of the methods are given in the earlier paper, which we will note in the table legend:

"*Table B1 Tidal marsh surface characteristics and soil characteristics for 0 - 40 cm depth including standard error. This table is presented as primary data in Pinsonneault et al. (2021) who present full descriptions of sample techniques and measurement methods.*"

**RW2:** The Instant Ocean salt origin (is it sea salt, or mineral salts mixed) and composition should be mentioned in the methods.

**Author:** Instant Ocean is a mineral salt mix and we will provide a citation for its composition:

"*The treated concentrate was divided into four sub-stocks that we amended with Instant Ocean aquarium salt (a synthetic sea salt) to produce four salinity treatments: 0 (no instant ocean added), 10, 20, and 35 on the practical salinity scale (no units). The typical ionic composition of Instant Ocean is reported by Christy and Dickman (2002).* "

**RW2:** pH measurements are mentioned, but I could not find the data in the results. Both, pH and Instant Ocean salt composition (specifically the divalent cations) should have an influence on adsorption-desorption properties of DOM. The data should be presented and discussed accordingly.

**Author:** As previously reported by Pinsonneault et al. (2020), pH increased in all incubations. A summary of their results will be provided in the discussion all with their implications:

*"Increasing pH also increases DOM absorbance (Gao et al., 2015), and postincubation pH increased in all our incubations from 4.6±0.09 in the standards to 6.74±0.03, 6.64±0.04, 5.15±0.05 and 6.94±0.10 for Kirkpatrick, Taskinas, Jug Bay and Wachapreague soils (respectively) (Pinsonneault et al., 2021). The increase in pH did not have an important effect on postincubation absorbance since specific absorbance decreased during the incubation, the opposite direction from what would be expected from the pH change alone."*

**RW2:** Assuming that DOM only interacts with leachable (poorly crystalline) iron and aluminum, it should be easy to evaluate whether the amount of leachable metals suffices to adsorb these huge amounts of concentrated DOM. There are plenty of experimental estimates on metals:carbon ratios and it would be an interesting calculation exercise in my view. Again, additional factors may play a role like divalent salt cations, arsenic or perhaps DOM aromatic-aromatic interactions.

**Author:** This is an interesting point that is mainly relevant to the net exchange of carbon with the soil which is the subject of our previous publication by Pinsonneault et al (2021). This previous report presents an in-depth discussion of the relationship of net sorption to leachable iron and aluminum content of the tidal marsh soils, we refer interested readers to that publication.

**Specific Comments:**

**RW2:** Section 3.3, line 281 "the spectral characteristics of DOC are a robust proxy…" is a bold statement considering the EEMs results in this manuscript. On the contrary, it seems that optical properties are limited to assessing CDOC behavior including absorbance and fluorescence. Consider rephrasing this and including a few sentences in the discussion on why there is so little additional insight.

**Author:** We agree that the statement is out of place, as also pointed out by Reviewer 1. As stated in the response to the other review, we will rephrase the sentence and move it to the second paragraph (see response to Reviewer 1).

**RW2:** Section 3.3, lines 292-293 "the average increase was slightly greater (…) than less for …" I think the "less" can be removed.

**Author:**  This was a typo and will be corrected:

"*Interestingly, the average increase was slightly greater for S=20 (0.017) and less for S=35 (0.014).*"

 **RW2:**  Acknowledgements, lines 490-491 "We would also like to thank the staff of the United States Department of Energy Environmental Molecular Science Laboratory for conducting the FT-ICR MS measurements." I'd be happy to see those data, but I guess the line is from another (future?) paper. Please conduct a final cross-check for typos and inconsistencies throughout the manuscript.

**Author:**  Good catch, we are not reporting FT-ICR measurements in this article, so this will be removed. Hopefully, we can provide those in the future.

Reference:

Christy, M. and Dickman, C.: Effects of salinity on tadpoles of the green and golden bell frog (Litoria aurea), Amphibia-Reptilia, 23, 1-11, 10.1163/156853802320877582, 2002.

---

## Author Response (AR1)

We thank the reviewers for their positive assessment of the work and detailed suggestions to improve the manuscript. We have carefully revised the text according to their comments, provide a marked-up version showing changes made, and list point-by-point responses to reviewer comments below. Line references in the responses are to the tracked version of the manuscript (Neale et al Sorption of Colored vs Noncolored OM R1 Tracked.pdf).

**Reviewer 1**

**Main Comments:**

**RW1:** The one major nagging question that I had throughout the entire review was that the authors have not accounted for, or even discussed anywhere, the potential limitations of using absorbance alone to partition colored versus non-colored DOC concentrations. Specifically, can shifts in absorbance in response to salinization induce optical changes in the DOM pool that alter absorbance through concentration-independent processes? Salt additions that alter ion binding patterns in the soil matrix could liberate/precipitate stuff like iron that may impact absorbance, no? It could also alter the chelation of DOM and metals and other materials, altering optical conditions? I am not raising these potential mechanisms to say that they invalidate the results. It is likely that at these DOC concentrations, such effects would be less important. BUT, using knowledge from the literature, and any other information on hand, can the authors speak to what role, if any, these effects have on their interpretations and conclusions (even if only to caveat them)?

**Author:** The reviewer raises two good points, one that solution salinity affects the optical properties of CDOM and secondly that salinity affects the binding to the soil matrix of other components, like iron, that may also affect absorbance properties. The first effect can be seen from the results in Table 6 which show slight decreases in CDOM slopes and slope ratios within each sample category as salinity is increased. Salinity increases are also linked to increased specific- absorbance at long wavelengths (Gao et al. 2015), but this effect contrasts with the observed trend in our post-incubation data of decreased specific absorbance (increased DOC slope, Table 1). As far as possible metal:DOC interactions in the GDS solution, the DOM was sampled from a freshwater peat marsh with very low soil iron – which is why DOM is high to begin with. Moreover, all absorbance measurements were made under anaerobic conditions. Thus dissolved iron, to the extent it was present, would mainly be in non-absorbing $Fe^{2+}$ form not the UV absorbing $Fe^{3+}$ form. We now include new text describing the absorbance methods (this text was inadvertently left out of our first submission) which notes that scans were done under anaerobic conditions (Line 130):

*"Due to the highly colored nature of the Great Dismal Swamp DOC, we diluted subsamples of pre-incubation standards by a factor of 10 using dilutant from the same salinity treatment while we diluted the filtered post-incubation supernatant by the same factor using ultrapure water + NaCl (Sigma Aldritch, 99.5% purity) to match sample salinity. All solutions were degassed with $N_2$, then dilutions were performed, cuvettes loaded and sealed in an anaerobic chamber. We*

*performed absorbance scans at 2 nm intervals (270 – 750 nm) for all replicates using a Thermo Scientific Evolution 220 UV-Vis spectrophotometer."*

In addition, we add comments on the effects of salinity and iron as well as pH change (a request from Reviewer 2) on our absorbance results to the discussion (Line 468):

*"Variations in incubation physiochemical conditions also affected DOM absorbance, but these variations do not substantially affect our interpretation of CDOC dynamics.  Increasing salinity, and thus ionic-strength, increases CDOM absorbance, with proportionally greater increases at longer wavelengths (Gao et al., 2015).  This is attributed to deprotonation of DOM, and results in a lowering of the spectral slope of CDOM absorption spectra and increase of specific absorbance, e.g. at 355 nm (Gao et al 2015).  Consistent with these effects is the slight reduction of spectral slope with increasing salinity in both pre- and post-incubation spectra, and the increase in specific absorbance (inverse of the decrease in DOC regression slope) of pre-incubation CDOC (Tables 1 and 6).  However, increasing salinity, per se, does not account for the trend in the key change related to desorption of NCDOC during incubations, the decrease in specific absorbance (increase in DOC slope). This decrease was intensified by increasing salinity (Tables 1).  Increasing pH also increases DOM absorbance (Gao et al., 2015), and pH increased in all our incubations from 4.6±0.09 in the standards to post-incubation values of 6.74±0.03, 6.64±0.04, 5.15±0.05 and 6.94±0.10 for Kirkpatrick, Taskinas, Jug Bay and Wachapreague soils (respectively) (Pinsonneault et al., 2021).  Again, the increase in pH did not have an important effect on post-incubation absorbance since specific absorbance decreased during the incubation, the opposite direction from what would be expected from the pH change alone.*

 *Ferric ($Fe^{3+}$) iron also absorbs in the UV and its concentration in some surface waters is high enough to significantly bias estimates of CDOM absorbance (Poulin et al., 2014; Logozzo et al., 2022). However, Fe is not a significant factor for DOM from the Great Dismal Swamp since it is a peat wetland with very low iron (USFWS, 2006).  Release of iron into solution during the incubation seems unlikely since it would have increased specific absorbance, again, the opposite of what we observed.  Finally, we can exclude interference from dissolved iron on our absorbance measurements since all solutions were anaerobic and dissolved iron would be in the non-absorbing $Fe^{2+}$ state."*

**Specific Comments:**

**RW1:** L23- GDS DOC – mention what this is at the start of abst. It's introduced in a strange spot.

**Author:** The source of the test solution is moved to the first sentence describing our experiments (Line 16):

*"To test this hypothesis, we generated initial mass sorption isotherms of CDOC and noncolored dissolved organic carbon (NCDOC) using anaerobic batch incubations of Great Dismal Swamp DOC with four tidal wetland soils, representing a range of organic carbon content (1.77 ± 0.12 % to 36.2 ± 2.2%) and across four salinity treatments (0, 10, 20, and 35)."*

**RW1:** L33 -  Intro – Polydisp. – define what this is in brackets.

**Author:** Definition is added to the sentence (underlined) (Line 33):

*"Studies have found that CDOM composition in flooding and ebbing tidal waters differs with more strongly colored and aromatic CDOM of higher average molecular weight and polydispersity [breadth of the molecular weight distribution] being exported from tidal marshes into estuarine waters during ebbing tide relative to that imported into marshes during flooding tide (Tzortziou et al., 2008; Tzortziou et al., 2011)."*

**RW1:** L58 – Run on sentence.

**Author:**  Not clear to which sentence the reviewer is referring.  We have checked the text in this part of the mss to ensure that we don't have a run on sentence.

**RW1:** L96 – What is a "filter cale"? Can you reword with something more generic? I have never seen the word "cale" before like this.

**Author:** This was a typo, corrected to "filter capsule" (Line 104)

**RW1:** L106- Is the NaN3 a preservative? Explain purpose briefly.

**Author:**  Description is added (Line 106):

*"The resulting filtered DOC concentrate was then treated with 1 mM sodium azide (NaN$_3$), a microbial inhibitor, …"*

**RW1:** L123- Any details about this regression R2/strength criteria used? Were relationships scrutinized in any way?

**Author:**  Regression $r^2$ values and parameter standard errors are presented in Table 1,  the $r^2$ values were all very near 1.

**RW1:** L163- Fig 1 caption. Explain blue solid/dashed arrows so readers don't need to sift through text to interpret figure.

**Author:**  Arrows are now identified in the caption:

*"Figure 1. Scatterplot of DOC content vs absorption coefficient at 355 nm in pre-incubation (triangles) and post-incubation (+ symbols) treatments in triplicate from the Kirkpatrick soil, S=10, isotherm experiment with fitted linear regressions.  The dashed arrows indicate the difference in DOC between the regression lines for selected post-incubations absorbances, the solid arrows identify the DOC of the pre-incubation line at these absorbances."*

**RW1:** L204 – Fig 3. I spent a bunch of time trying to figure this out, then realized you are using negative values to mean desorption. Please state this up front in the caption. Even better would be to add a dashed zero line to signify the difference.

**Author:** The figures did have a zero gray line, this is now a dashed line that is more visible.  Also, we note the meaning of negative values in the figure caption:

*"Figure 3. Isotherms of adsorption-desorption of CDOC (filled shapes) and NCDOC (open shapes) for tidal marsh soils. Net adsorption of organic carbon based on the difference between initial and final concentrations of soil organic carbon ($\Delta C$, mg C g-soil-1) as a function of initial concentration of CDOC ([CDOC]i, mg L-1). Dashed line indicates no net change and negative values indicate net desorption. Error bars show standard deviation of triplicate incubations, in most cases error bars are smaller than the symbol. Incubations used soils from (a) Kirkpatrick, (b) Taskinas, (c) Jug Bay, and (d) Wachapreague marshes. Red circles, green triangles, blue squares, and purple diamonds denote salinities of 0, 10, 20, and 35, respectively."*

**RW1:** L215- All of this isotherm theory and mathematical model interpretation would be far easier to follow as a section fully developed in the methods, perhaps with a conceptual diagram of the isotherm graph.

**Author:**  The text describing the linear desorption isotherm is moved to the methods (line 173). Figure 4 was included to assist the reader in visualizing the context for the different isotherm models (see comment below).

**RW1:** L240- Throughout results, the ability for readers to wade through this info would be improved with the use of sub-headings to group info by themes.

**Author:** We added more sub-headings in the results,  3.2 CDOC Sorption Isotherms, 3.3 NCDOC Desorption Isotherms and 3.4 Relationship of Isotherm Parameters to Soil Characteristics

**RW1:** L251- Figure 4- Confusing. Is the y axis still initial minus final? If so, make that one clear.

**Author:** To further emphasize the occurrence of net desorption for some initial concentrations of CDOC a dashed line has been added at zero. This is now described in the figure caption:

*"Figure 4. Comparison of sorption isotherms for incubation of DOC standards with soil from Kirkpatrick Marsh, S=10, for TDOC (x, dashed line), CDOC (solid circles and line), and NCDOC (unfilled circles and solid line). Net adsorption of organic carbon, based on the difference between initial and final concentrations of soil carbon ($\Delta C$, mg C g-soil$^{-1}$), is plotted vs solution TDOC (for Total) or CDOC (Colored and Noncolored). Points show measured/estimated quantities from triplicate incubations for each standard, lines show the fitted Langmuir initial mass isotherm (TDOC,CDOC) and linear desorption isotherm (NCDOC). Dash-dot line indicates no net change, negative values indicate net desorption."*

**RW1:** L273- Again, a bunch of this background definition content would be nice to see in a dedicated methods section up front.

**Author:** The definition of null point and description of its meaning is moved to methods (Line 166):

*"The null point (NP) is defined as the concentration at which there is no net removal of CDOC from solution (adsorption) or release from soil (desorption), the biogeochemical significance of which is a sorptive equilibrium between the soil mineral surfaces and the aqueous phase. It is derived by setting Eq 2 to zero and solving for NP concentration of [CDOC]$_i$:*

$$NP = \frac{C_{C0}}{(Q_{Cmax} - C_{C0}) * K_c}$$
(3)"

**RW1:** L281- Initial sentence not needed.

Author: We agree that it was not needed for this paragraph. The purpose was to provide a segue to the spectral results, which is better accomplished by having a rephrased version as the topic sentence of the following paragraph (Line 334):

*"Given that spectral characteristics of CDOM absorbance and fluorescence are used to infer sources, sinks, reactivity, and other biogeochemical processes (Chin et al., 2002; Helms et al., 2008; Tzortziou et al., 2008), we examined how spectral properties of CDOC were affected by sorption processes."*

**RW1:** L293- "than less" – grammar.

**Author:** Changed to "and less" (Line 341)

**RW1:** L330- Do the authors find this odd that these components decrease? NCDOC increases, so should these components not also go up? I am not saying that they must, I am likely missing something here, but just want the authors to explain this to me/readers.

**Author:** It is not clear to which component decrease the reviewer is referring. Line 333 spoke of the decrease in the phenylalanine component, C4. As something prevalent in the GDS, it is regarded as a component of the CDOC, not the NCDOC, so there is no inconsistency with the increase in NCDOC.

**RW1:** L345- This figure feels a bit unnecessary/of limited use in current format. I'd swap out the text "dismalswamp" with something generic so that someone outside the region will understand is the stock solution without knowing the location it came from, and not having to consult the methods again. The grey/brown DOC could be defined in a legend so it is visually obvious what the color coding means. The figure could be redone to reflect what increasing salinity does to DOC sorption, which is a key question as I take it in the paper.

**Author:** Based on the reviewer's helpful comments the figure has been revised:

[Figure]

The different phases are now described in general terms and the color code for CDOC and NCDOC is explicit in the diagram.  Separate panels depict the increase in net adsorption of CDOC and desorption of NCDOC at high salinity.

**RW1:** L362- Ref format in brackets, run on sentence as written.

**Author:**  The text is broken up into two sentences (Line 428):

*"However, this was a minor process under our experimental conditions because incubations had minimal effects on the slopes and slope ratios of absorbance spectra.   Moreover, such differential adsorption would be expected to cause differential changes in the specific absorbance of CDOC (an indicator of average molecular weight) as a function of CDOC (cf. adsorption of fulvic acid on goethite Zhou et al., 2001) whereas we observed uniform specific absorbance over all incubation conditions (Fig. 1)."*

**RW1:** L364- Why redefining GDS here?

**Author:**  Abbreviation redefinition deleted. (Line 433)

**RW1:** L393- Photodegradation reference needed here.

**Author:**  Citation added to Dainard et al (2015) (Line 490).

**RW1:** L420- If the conceptual figure could link to this sentence it would be more useful. See my comment above about incorporating salinity.

**Author:**  The figure is now modified to illustrate our interpretation of the salinity effect and reference to figure is added to the cited sentence (Line 522).

**RW1:** L437- You come back to flood/ebb differences in DOM here. A few more sentences integrating the mechanisms exposed by the experiment, and how they potentially help to explain the empirical observations in the literature, would strengthen the discussion.

**Author:** We added new text placing sorption processes in the context of flood/ebb differences (line 539):

*"Porewater draining from tidal creek banks at low-tide is a primary source of CDOM in the ebbing tide (Menendez et al., 2022). Our results are consistent with this CDOM being derived by exchange with marsh sediment, as low DOC flood waters infiltrate into the sediment and desorb CDOC from the sediment.  Our results also suggest that sediments are a significant source of NCDOC which, for example, comprises ~50% of the DOC in Kirkpatrick Marsh tidal creeks at low tide"*

**Reviewer 2**

**Main Comments:**

**RW2:** The methods lack information on precision and trueness of TOC, DOC and metals measurements. EEMs are introduced in the results only. I understand the reasoning, but it still comes as a bit of a surprise and lacks important information on measurements. This should be more detailed and go into the methods section. In my view the rationale can be mentioned without giving away too much of the results.

**Author:** We inadvertently omitted the methods description for absorbance and fluorescence which is now provided (Line 126):

*"Due to the highly colored nature of the Great Dismal Swamp DOC, we diluted subsamples of pre-incubation standards by a factor of 10 using dilutant from the same salinity treatment while we diluted the filtered post-incubation supernatant by the same factor using ultrapure water + NaCl (Sigma Aldritch, 99.5% purity) to match sample salinity. All solutions were degassed with $N_2$, then dilutions were performed, cuvettes loaded and sealed in an anaerobic chamber.  We performed absorbance scans at 2 nm intervals (270 – 750 nm) for all replicates using a Thermo Scientific Evolution 220 UV-Vis spectrophotometer. Specific ultraviolet absorbance at 280 nm ($SUVA_{280}$), an indicator of DOC aromaticity, was calculated from this data by dividing decadic sample absorbance at 280 nm by the DOC concentration ([DOC]) (Hansen et al., 2016). We then generated three-dimensional excitation-emission matrices (EEMs) using a Horiba Jobin Yvon FluoroMax-3 spectrofluorometer (for sample replicate A only) at 5 nm intervals (250 – 600 nm) for excitation and 2 nm intervals (250 – 600 nm) for emission. Fluorescence spectra were corrected for inner-filter effect and Raman scattering using the drEEM toolbox version 0.2.0 (Murphy et al. 2013) in MATLAB (v. 2017b). Parallel factor analysis (PARAFAC) was used to deconstruct the fluorescence signal into underlying fluorescence components, or fluorophores, that relate to differences in DOC composition (Murphy et al., 2010; Lapierre and del Giorgio, 2014)."*

Metal measurements are reprinted in Appendix Table B1 from Pinsonneault et al. (2021).  Since all the methods and this data were reported in the earlier paper, we now omit description of the soils methods from this report.  This is mentioned in the table legend:

"*Table B1 Tidal marsh surface characteristics and soil characteristics for 0 - 40 cm depth including standard error. This table is presented as primary data in Pinsonneault et al. (2021) who present full descriptions of sample techniques and measurement methods.*"

**RW2:** The Instant Ocean salt origin (is it sea salt, or mineral salts mixed) and composition should be mentioned in the methods.

**Author:** Instant Ocean is a mineral salt mix and we provide a citation for its composition (Line 110):

"*The treated concentrate was divided into four sub-stocks that we amended with Instant Ocean aquarium salt (a synthetic sea salt) to produce four salinity treatments: 0 (no instant ocean added), 10, 20, and 35 on the practical salinity scale (no units). The typical ionic composition of Instant Ocean is reported by Christy and Dickman (2002).* "

**RW2:** pH measurements are mentioned, but I could not find the data in the results. Both, pH and Instant Ocean salt composition (specifically the divalent cations) should have an influence on adsorption-desorption properties of DOM. The data should be presented and discussed accordingly.

**Author:**  As previously reported by Pinsonneault et al. (2020), pH increased in all incubations. A summary of their results is provided in the discussion with their implications (Line 476):

"*Increasing pH also increases DOM absorbance (Gao et al., 2015), and pH increased in all our incubations from 4.6±0.09 in the standards to post-incubation values of 6.74±0.03, 6.64±0.04, 5.15±0.05 and 6.94±0.10 for Kirkpatrick, Taskinas, Jug Bay and Wachapreague soils (respectively) (Pinsonneault et al., 2021).  Again, the increase in pH did not have an important effect on post-incubation absorbance since specific absorbance decreased during the incubation, the opposite direction from what would be expected from the pH change alone.*"

**RW2:** Assuming that DOM only interacts with leachable (poorly crystalline) iron and aluminum, it should be easy to evaluate whether the amount of leachable metals suffices to adsorb these huge amounts of concentrated DOM. There are plenty of experimental estimates on metals:carbon ratios and it would be an interesting calculation exercise in my view. Again, additional factors may play a role like divalent salt cations, arsenic or perhaps DOM aromatic-aromatic interactions.

**Author:** This is an interesting point that is mainly relevant to the net exchange of carbon with the soil which is the subject of our previous publication by Pinsonneault et al (2021).  This previous report presents an in-depth discussion of the relationship of net sorption to leachable

iron and aluminum content of the tidal marsh soils, we refer interested readers to that publication.

**Specific Comments:**

**RW2:** Section 3.3, line 281 "the spectral characteristics of DOC are a robust proxy…" is a bold statement considering the EEMs results in this manuscript. On the contrary, it seems that optical properties are limited to assessing CDOC behavior including absorbance and fluorescence. Consider rephrasing this and including a few sentences in the discussion on why there is so little additional insight.

**Author:** We agree that the statement is out of place, as also pointed out by Reviewer 1. As stated in the response to the other review, we have rephrased the sentence and move it to the second paragraph starting on Line 334 (see response to Reviewer 1).

**RW2:** Section 3.3, lines 292-293 "the average increase was slightly greater (…) than less for …" I think the "less" can be removed.

**Author:** This was a typo and is corrected (Line 341):

"*Interestingly, the average increase was slightly greater for S=20 (0.017) and less for S=35 (0.014).*"

**RW2:** Acknowledgements, lines 490-491 "We would also like to thank the staff of the United States Department of Energy Environmental Molecular Science Laboratory for conducting the FT-ICR MS measurements." I'd be happy to see those data, but I guess the line is from another (future?) paper. Please conduct a final cross-check for typos and inconsistencies throughout the manuscript.

**Author:** Good catch, we are not reporting FT-ICR measurements in this article, so this is removed. Hopefully, we can provide those data in the future.